# A class I PI3K signalling network regulates primary cilia disassembly in normal physiology and disease

Sarah E. Conduit [1] ✉, Wayne Pearce[1], Amandeep Bhamra[2], Benoit Bilanges[1], Laura Bozal-Basterra [3,4], Lazaros C. Foukas [5], Mathias Cobbaut [6], Sandra D. Castillo [7], Mohammad Amin Danesh[1], Mahreen Adil[1], Arkaitz Carracedo [3,4,8,9,10], Mariona Graupera [4,7,11], Neil Q. McDonald [6,12], Peter J. Parker [13,14], Pedro R. Cutillas [15], Silvia Surinova [2] & Bart Vanhaesebroeck [1] ✉

Primary cilia are antenna-like organelles which sense extracellular cues and act as signalling hubs. Cilia dysfunction causes a heterogeneous group of disorders known as ciliopathy syndromes affecting most organs. Cilia disassembly, the process by which cells lose their cilium, is poorly understood but frequently observed in disease and upon cell transformation. Here, we uncover a role for the PI3Kα signalling enzyme in cilia disassembly. Genetic PI3Kα-hyperactivation, as observed in *PIK3CA*-related overgrowth spectrum (PROS) and cancer, induced a ciliopathy-like phenotype during mouse development. Mechanistically, PI3Kα and PI3Kβ produce the $PIP_3$ lipid at the cilia transition zone upon disassembly stimulation. PI3Kα activation initiates cilia disassembly through a kinase signalling axis via the PDK1/PKCι kinases, the CEP170 centrosomal protein and the KIF2A microtubule-depolymerising kinesin. Our data suggest diseases caused by PI3Kα-activation may be considered 'Disorders with Ciliary Contributions', a recently-defined subset of ciliopathies in which some, but not all, of the clinical manifestations result from cilia dysfunction.

Primary cilia are sensory organelles which project from the surface of most mammalian cell types. The protrusion of cilia into the extracellular environment and enrichment of receptors at this organelle enable it to act as the cell's antenna for developmental and homeostatic signals including Hedgehog, Wnt, receptor tyrosine kinase (RTK) and G protein-coupled receptor (GPCR) signalling[1]. The critical requirement for a functional cilium is highlighted by the broad phenotypic spectrum of ciliopathies resulting from

[1]Cell Signalling, UCL Cancer Institute, University College London, 72 Huntley Street, London WC1E 6BT, UK. [2]Proteomics Research Translational Technology Platform, UCL Cancer Institute, University College London, 72 Huntley Street, London WC1E 6BT, UK. [3]Center for Cooperative Research in Biosciences (CIC bioGUNE), Basque Research and Technology Alliance (BRTA), Bizkaia Technology Park, Building 801A, 48160 Derio, Spain. [4]Centro de Investigación Biomédica En Red de Cáncer (CIBERONC), 28029 Madrid, Spain. [5]Institute of Healthy Ageing, Department of Genetics, Evolution and Environment, University College London, London WC1E 6BT, UK. [6]Signalling and Structural Biology laboratory, The Francis Crick Institute, 1 Midland Road, London NW1 1AT, UK. [7]Endothelial Pathobiology and Microenvironment, Josep Carreras Leukaemia Research Institute, Barcelona, Spain. [8]Translational Prostate Cancer Research Laboratory, CIC bioGUNE-Basurto, Biocruces Bizkaia Health Research Institute, Barakaldo, Spain. [9]IKERBASQUE, Basque Foundation for Science, 48009 Bilbao, Spain. [10]Biochemistry and Molecular Biology Department, University of the Basque Country (UPV/EHU), P.O. Box 644, E-48080 Bilbao, Spain. [11]ICREA, Institució Catalana de Recerca i Estudis Avançats, Pg. Lluís Companys 23, Barcelona, Spain. [12]Institute of Structural and Molecular Biology, School of Natural Sciences, Birkbeck College, Malet Street, London WC1E 7HX, UK. [13]The Francis Crick Institute, 1 Midland Road, London NW1 1AT, UK. [14]King's College London, Guy's Campus, London, UK. [15]Centre for Genomics and Computational Biology, Barts Cancer Institute, Queen Mary University of London, London EC1M 6BQ, UK. ✉e-mail: s.conduit@ucl.ac.uk; bart.vanh@ucl.ac.uk

loss-of-cilia-function mutations, including neurodevelopmental abnormalities, renal and hepatic cysts, polydactyly, retinal degeneration and frequently embryonic lethality[2].

Lovera and coworkers recently coined the terminology 'disorders with ciliary contribution' (DCC) as the subset of ciliopathies resulting from mutations in genes with both ciliary and non-ciliary functions, where the latter may mask the clinical cilia dysfunction phenotypes[3]. In these cases, as well as many established ciliopathies, the mutant proteins frequently exhibit both ciliary and non-ciliary localisations and functions, a recent change in direction of the field that has broadened the discovery of proteins that contribute to cilia biology and associated disease states[4]. These observations have led to the suggested distinction of first- and second-order ciliopathies, whereby first-order ciliopathies are caused by mutations in proteins that localise to cilia or centrosomes, with second-order ciliopathy proteins not localising to these sites[3]. Cilia dysfunction is also emerging in non-syndromic conditions such as neurodegeneration[5,6], inflammation[7] and cancer[8].

Primary cilia are dynamic, and their presence is dictated by an equilibrium between cilia assembly, maintenance and disassembly[9]. Defects in these processes underly some ciliopathies[9,10]. Furthermore, most solid tumour types exhibit a loss of ciliated cells compared to their corresponding non-transformed cell states[11–15]. While the molecular mechanisms of cilia assembly are well-characterised, the process of cilia disassembly and how cancer cells lose cilia remains poorly understood[10]. While some proteins have been identified to induce cilia disassembly in individual, candidate-focused studies[16–21], how these proteins interact and function in a coordinated manner in response to cilia disassembly stimuli remains unclear.

Emerging evidence suggests that phosphoinositide (PI) lipids in the cilia membrane are central in cilia disassembly[22,23]. PI lipids define membrane identity and are central regulators of cell signalling[24,25]. Interestingly, the ciliary membrane and lumen exhibit a distinct protein and phospholipid composition from the plasma membrane and cytosol, despite being directly interconnected. This remarkable separation is controlled by the barrier function of the so-called transition zone at the cilia base. Four PI species decorate the ciliary membrane. PI(4)P defines the axoneme membrane[26,27], whereas $PI(4,5)P_2$, $PI(3,4)P_2$ and $PI(3,4,5)P_3$ (the latter hereafter referred to as $PIP_3$) are enriched at the transition zone membrane[23,26–28]. The best studied ciliary PIs are PI(4)P and $PI(4,5)P_2$ in the axoneme membrane which control exocytosis and ciliary localisation of GPCRs and are regulated by the PI 5-phosphatase INPP5E[22,26,27,29].

In contrast to PI(4)P and $PI(4,5)P_2$, the regulation and function of $PIP_3$ and $PI(3,4)P_2$ at cilia is poorly understood. The latter lipids are universally dysregulated in cancer and also implicated in overgrowth syndromes, and mainly produced by the PI3K enzymes. Using super-resolution microscopy, we showed that $PIP_3$ localises in a previously-unappreciated ring-shaped sub-domain at the cilia transition zone, distal to $PI(4,5)P_2$[28,30]. $PIP_3$ levels at this site increase in response to stimulation with the cilia disassembly inducer IGF-1[23]. $PIP_3$ is also hydrolysed by the INPP5E phosphatase, with increased ciliary $PIP_3$ in *Inpp5e* knockout cells associated with defective transition zone barrier function and induction of cilia disassembly[23,28]. Although a role for the PI3K effector AKT and its target GSK3β in cilia disassembly has been proposed[23], this has not been experimentally validated and the signalling mechanisms by which $PIP_3$ regulates cilia disassembly remain to be elucidated.

Plasma membrane $PIP_3$ is generated by the class I PI3Ks (PI3Kα, β, γ and δ) which are activated downstream of RTKs, GPCRs and small GTPases. Class I PI3Ks are activated by serum, LPA and growth factors including IGF-1[31,32], all of which are cilia disassembly inducers[18,33,34]. However, there is no evidence of a regulated production of these PIs controlling cilia dynamics and the PI3K isoform(s) responsible for production of ciliary $PIP_3$ in basal and stimulated conditions remains elusive. Furthermore, we and others previously showed that the

pharmacological kinase inhibitor LY294002 inhibits cilia disassembly[23,35], however, this compound inhibits all eight PI3K isoforms as well as a range of additional enzymes[36,37]. Many solid tumour cells and cell lines exhibit reduced ciliated cells compared to their non-transformed counterparts[11–15] and studies propose that cilia loss promotes transformation[38–40]. Class I PI3K signalling is one of the most frequently genetically activated kinase pathways in cancer.

Here, we show that the ubiquitously-expressed PI3Kα and PI3Kβ class I PI3K isoforms, and an associated signalling network, regulate primary cilia disassembly in cells, development and cancer, and speculate that diseases caused by aberrant PI3Kα-activation may be considered as a DCC, a subset of ciliopathies in which some of the clinical manifestations result from cilia dysfunction.

## Results

### Constitutive in vivo activation of PI3Kα induces ciliopathy phenotypes and repression of cilia-dependent signalling

PI3K overactivation in cancer is most often due to activating mutations in *PIK3CA*, the gene encoding the PI3Kα catalytic subunit. Similar *PIK3CA* mutations also cause the rare overgrowth disorder called *PIK3CA*-related overgrowth spectrum (PROS). To explore the role of oncogenic *PIK3CA* in cilia biology, we re-examined a mouse model with heterozygous ubiquitous constitutive PI3Kα hyperactivation. Specifically, $Pik3ca^{tm1.1Waph/+}$;$Tg(CMV\text{-}cre)1Cgn$ (hereafter referred to as $Pik3ca^{H1047R}$) mice in which zygotically-expressed CMV-Cre drives expression of the constitutively active oncogenic $Pik3ca^{H1047R}$ allele from the endogenous *Pik3ca* promoter, which are embryonically lethal at E9.5 (Ref. 41 and Supplementary Table 1).

Previous characterisation of these $Pik3ca^{H1047R}$ embryos focused on the defective angiogenesis and vascular remodelling phenotypes[42]. Notably however, we observed that these embryos also exhibit reduced size and somite number, shorter posterior trunk, pale colour, failed neural tube closure and defective turning (Fig. 1a, Supplementary Table 2)[42]. These embryonic patterning phenotypes are characteristic of primary cilia and cilia-signalling (Hedgehog and canonical Wnt) mutant mouse models (Supplementary Data 1) but, somewhat remarkably, have to date not been further investigated in $Pik3ca^{H1047R}$ embryos. Importantly, endothelial-specific $Pik3ca^{H1047R}$ expression results in embryonic lethality later than E9.5 and does not perturb turning or patterning[42], indicating that the cilia-associated phenotypes in $Pik3ca^{H1047R}$ mice are a primary effect of $Pik3ca^{H1047R}$ expression, and not a consequence of the angiogenesis defects.

We next performed a more targeted phenotypic characterisation with a focus on cilia dysfunction. The high cell density in embryonic tissues makes it difficult to score ciliated cells[43]. We therefore used mouse retinal endothelial cells which exhibit primary cilia, with the proportion of ciliated cells peaking in early postnatal mice[44]. Specifically, we assessed the proportion of ciliated endothelial cells in neonatal retinas from $Pdgfb\text{-}CreER^{T2}$;$Pik3ca^{H1047R}$ mice which heterozygously express $Pik3ca^{H1047R}$ in endothelial cells following 4-hydroxytamoxifen-induced recombination at postnatal day 1 (P1) and exhibit vascular malformations by P6[45]. As can be seen from Fig. 1b, the proportion of ciliated endothelial cells was subtly but significantly reduced in $Pdgfb\text{-}CreER^{T2}$; $Pik3ca^{H1047R}$ retinal endothelial cells compared to controls.

Given the $Pik3ca^{H1047R}$ cilia related-phenotypes resemble cilia-dependent Hedgehog and canonical WNT signalling loss-of-function mutant mice (Supplementary Data 1), we next measured the activity of these pathways in $Pik3ca^{H1047R}$ embryos by quantifying the mRNA levels of 'gold-standard' target genes[46–49]. It is well-established that loss of primary cilia represses Hedgehog signalling in the whole embryo[50], but can activate the pathway in specific tissues more dependent on GLI3 repressor processing, such as the limb bud and craniofacial primordium[51,52]. As shown in Fig. 1c, d and Supplementary Fig. 1a, the mRNA levels of Hedgehog target genes *Gli1* and *Ptch1* were reduced in

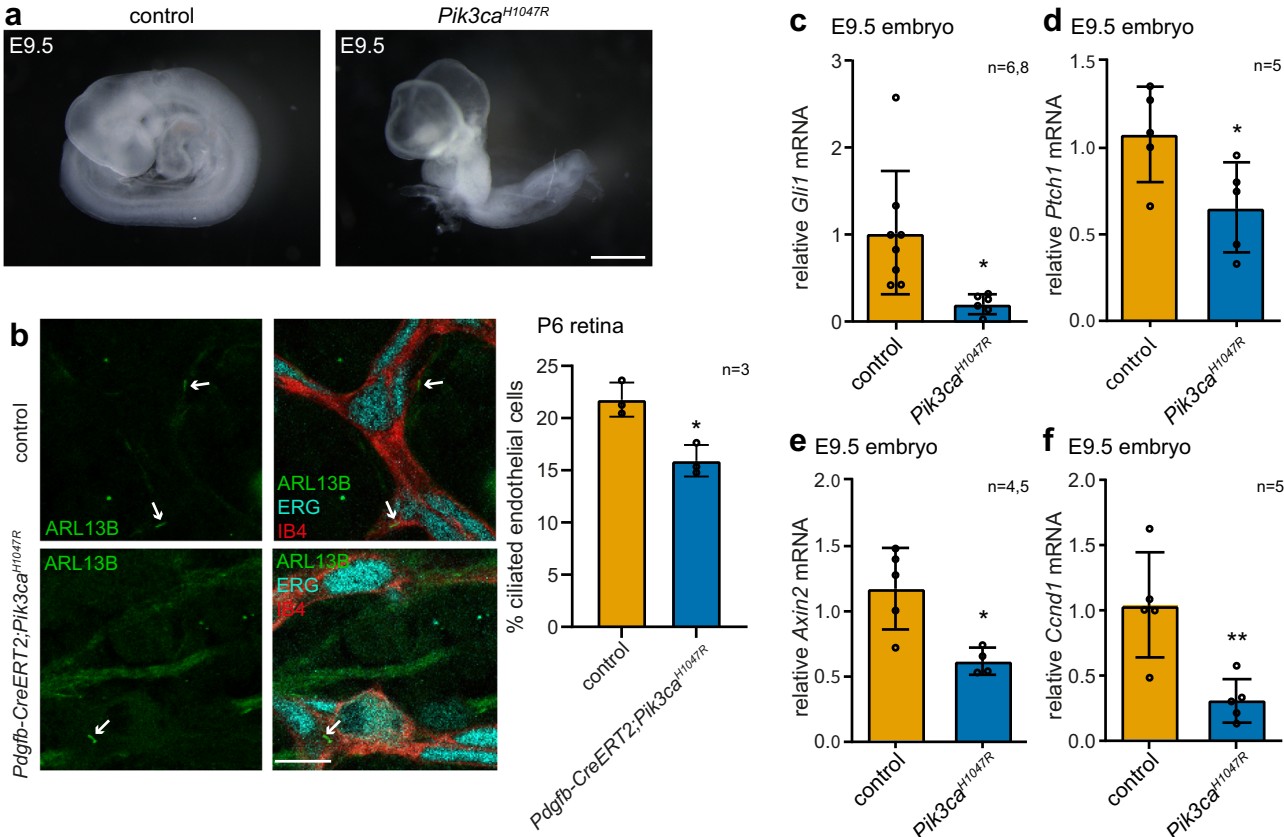

**Fig. 1 | Constitutive activation of PI3Kα in vivo induces ciliopathy phenotypes and repression of cilia-dependent signalling. a** Whole mount images of E9.5 control and *Pik3ca^H1047R* embryos, scale bar: 500 μm, representative of *n* = 59 (control), *n* = 48 (*Pik3ca^H1047R*) embryos. **b** Retinas from *Pik3ca^H1047R* and *Pdgfb-CreER^T2;Pik3ca^H1047R* mice treated with 4-hydroxytamoxifen at P1 were harvested at P6, immunostained with ARL13B, IB4 and ERG antibodies and imaged by confocal microscopy, scale bar: 10 um. The proportion of ciliated endothelial cells was scored, bars represent mean ± SD, ≥397 cells were scored per mouse for *n* = 3 mice/ genotype, *\*p* < 0.05 (two-sided Student's *t*-test, *p* = 0.0104). E9.5 control and *Pik3ca^H1047R* embryos were lysed and (**c**) *Gli1* or (**d**) *Ptch1* mRNA levels quantified by qRT-PCR relative to *Gapdh*, bars represent mean ± SD, (**c**) *n* = 6,8, (**d**) *n* = 5 mice of each genotype, *\*p* < 0.05 (two-sided Student's *t*-test, (**c**) *p* = 0.0130, (**d**) p = 0.0365). E9.5 control and *Pik3ca^H1047R* embryos were lysed and (**e**) *Axin2* or (**f**) *Ccnd1* mRNA levels quantified by qRT-PCR relative to *Gapdh*, bars represent mean ± SD, (**e**) *n* = 4,5, (**f**) *n* = 5 mice of each genotype, *\*p* < 0.05, **\**p* < 0.01 (two-sided Student's *t*-test, (**e**) *p* = 0.0113, (**f**) *p* = 0.0059). Source data are provided as a Source Data file.

E9.5 *Pik3ca^H1047R* embryos relative to littermate controls, and a trend for repression was observed at E9.

The relationship between cilia and canonical Wnt signalling in mice is complex, possibly dependent on the tissue and genetic background. Some studies show cilia dysfunction increases canonical Wnt signalling[53–59], others show no change in signalling[60] and analysis of cilia dysfunction mutants on a congenic (>10 backcrossed generations) C57BL/6 background identified repression of canonical Wnt signalling[55,61,62]. We found that E9 and E9.5 *Pik3ca^H1047R* congenic C57BL/6 embryos showed reduced expression of the WNT target genes *Axin2* and *Ccnd1* relative to controls (Fig. 1e, f, Supplementary Fig. 1b), consistent with characteristic WNT-dependent elements of the phenotype such as caudal body truncation.

To assess whether repression of Hedgehog signalling upon *Pik3ca^H1047R* expression is a cell-autonomous effect, we quantified the response to SAG, an agonist of the Smoothened GPCR in Hedgehog-responsive human hTERT-RPE1 retinal pigment epithelial cells expressing *PIK3CA^H1047R*. To this end, we generated hTERT-RPE1 cells inducibly expressing *PIK3CA^H1047R* under the control of doxycycline (DOX). Hyperactivation of PI3K/AKT/pS6RP signalling was validated in both ciliated and non-ciliated *PIK3CA^H1047R* hTERT-RPE1 cells relative to vector controls (Supplementary Fig. 1c, d) and *PIK3CA^H1047R*-expressing cells exhibited a reduced percentage of ciliated cells upon serum starvation (Fig. 1e). As expected, SAG induced robust *GLI1* expression in vector control cells, however, this transcriptional response was not

observed upon *PIK3CA^H1047R* expression (Supplementary Fig. 1f), confirming cell-autonomous repression of Hedgehog signalling upon PI3Kα activation. SMO is known to accumulate at the primary cilium in response to Hedgehog pathway activation, an event critical for downstream signalling but abrogated in the context of high ciliary PIP₃ levels in *Inpp5e* null cells[28]. Therefore, SMO ciliary localisation was examined in cilia remaining in *PIK3CA^H1047R*-expressing hTERT-RPE1 cells stimulated ± SAG for 24 h, revealing a robust axonemal SMO accumulation in SAG-treated vector control cells, which was impaired by *PIK3CA^H1047R* expression (Supplementary Fig. 1g). Indeed, reduced ciliary SMO in remaining ciliated cells would contribute to the repression of Hedgehog signalling, however, a reduced population of ciliated cells capable of transducing the signal will also contribute and is likely to have a larger impact.

## Activation of class I PI3K/AKT, MAPK, PKC and PKA during cilia disassembly

In vitro, the process of cilia assembly occurs over a 2-day serum deprivation period. Cilia disassembly can then be induced by serum or LPA stimulation over a 24 h period (Fig. 2a)[18,34]. We hypothesised the cilia dependent phenotypes observed in *Pik3ca^H1047R*-mice may therefore be a consequence of deregulated cilia disassembly.

Serum and LPA activate pathways such as PI3K, MAPK, PKC and PKA which are typically experimentally assessed in cells starved for short time periods (hours) under conditions in which these cells do

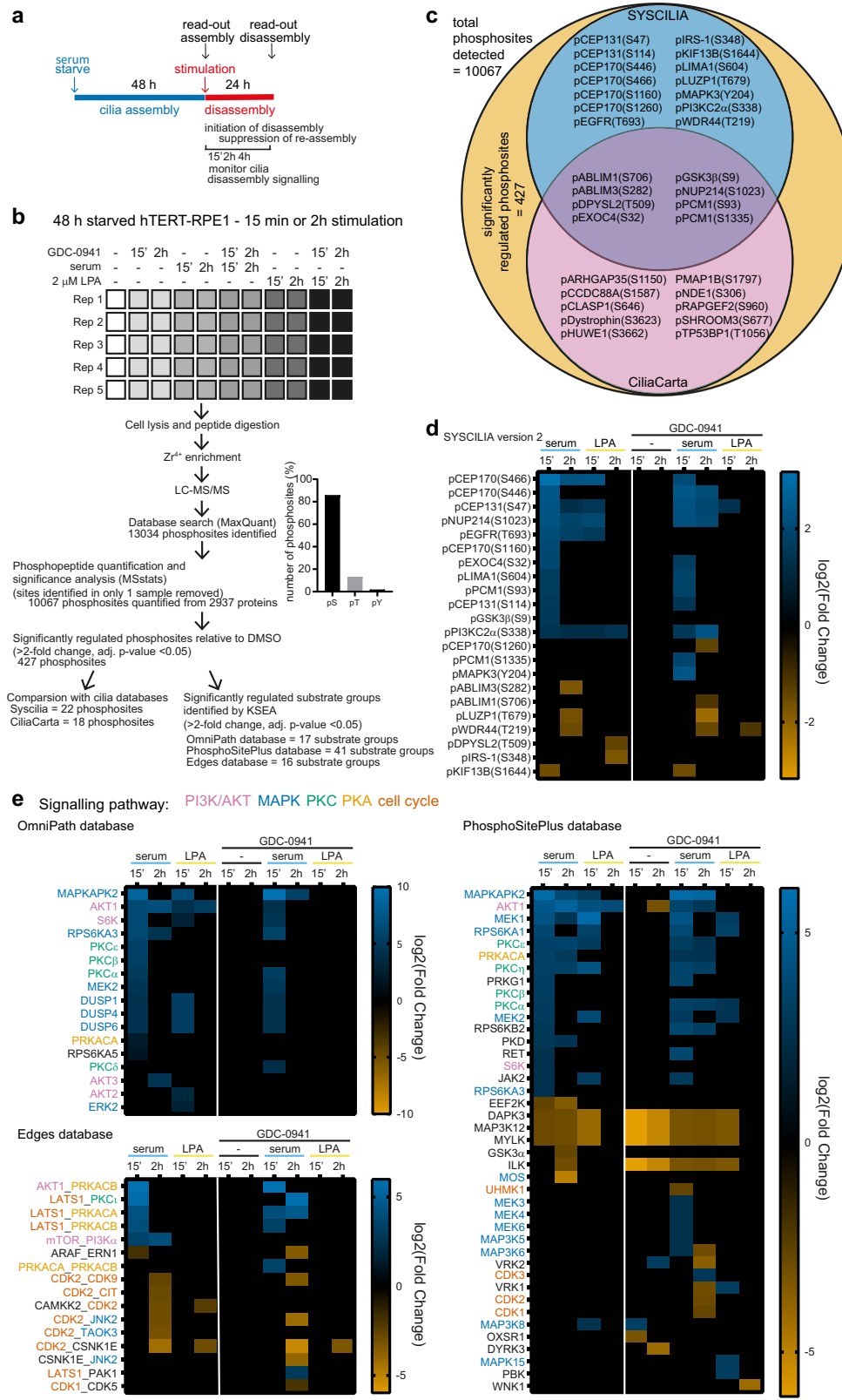

not express cilia. Surprisingly, the signalling of these kinases has not been established under experimental conditions of cilia disassembly. We therefore first asked which kinases are active in ciliated cells undergoing cilia disassembly. For this, we used untargeted phosphoproteomics and kinase substrate enrichment analysis (KSEA)[63] of the well-established hTERT-RPE1 cilia cell model, in which 80% of cells are ciliated following 48 h serum starvation[34]. Whole cell lysates

were used, rather than fractionated cilia, given that (1) cilia isolation methods often exclude the transition zone and basal body[64–67]; (2) the field has thus far not managed to perform phospho-enrichment on isolated cilia extracts given the current technical limitations of low sample input phosphoproteomics; and (3) whole cell lysate proteomic approaches have in the past allowed the discovery of relevant cilia biology[68].

**Fig. 2 | Cilia disassembly stimuli activate PI3K/AKT, MAPK, PKC and PKA signalling. a** Layout of experimental induction and analysis of serum starvation-induced cilia assembly and stimulation-induced cilia disassembly. **b** Experimental design and workflow of phosphoproteomic experiment in hTERT-RPE1 cells. Cells were serum-starved for 48 h and stimulated with 10% serum or 2 µM LPA ± 0.5 µM GDC-0941 for 15 min or 2 h and processed for phosphoproteomic analysis. 10067 phosphosites from 2937 proteins were analysed by MSstats, 427 phosphosites were significantly regulated which were used for comparison with cilia proteome databases and for KSEA (*n* = 5 independent experiments). **c** Venn diagram showing phosphosites regulated by serum or 2 µM LPA ± 0.5 µM GDC-0941 of 48 h serum-starved hTERT-RPE1 cells relative to DMSO. 10067 phosphosites were quantified by phosphoproteomics of which 427 were differentially regulated. Phosphosites in

cilia-associated proteins (as defined by SYSCILIA version 2 or CiliaCarta) are listed. **d** Heatmap displaying phosphosites from cilia-associated proteins (as defined by SYSCILIA) regulated by 15 min or 2 h serum or 2 µM LPA stimulation ± 0.5 µM GDC-0941 in 48 h serum-starved hTERT-RPE1 cells relative to DMSO. **e** Heatmaps displaying KSEA (using the OmniPath, Edges and PhosphoSitePlus database) of kinases for which the substrate groups were differentially regulated by 15 min or 2 h serum or 2 µM LPA stimulation ± 0.5 µM GDC-0941 in 48 h serum-starved hTERT-RPE1 cells relative to DMSO. Kinases for which the adjusted *p*-values (using the Kolmogorov–Smirnov test, adjusted for multiple comparisons with Benjamini-Hochberg principle (5% FDR)) relative to DMSO control were less than *p* = 0.05 were considered significantly regulated.

Starved ciliated hTERT-RPE1 cells were treated with serum or LPA for 15 min or 2 h, with or without GDC-0941 to test the involvement of class I PI3Ks (Fig. 2b). Phosphosites exhibiting >2-fold change relative to DMSO and adjusted *p*-value < 0.05 were defined as significantly regulated. 10,067 phosphosites (Supplementary Data 2) from 2937 proteins were quantified, including 8643 pSer, 1332 pThr and 92 pTyr residues (Fig. 2b). 427 phosphosites were significantly regulated by these treatments, with the majority upregulated by serum or to a lesser extent LPA, and a subset partially normalised by GDC-0941 (Supplementary Fig. 2b). 32 phosphosites were in cilia-associated proteins as defined by SYSCILIA (version 2)[69] and CiliaCarta[70] (Fig. 2c, d; Supplementary Fig. 2c). The phosphorylation of multiple cilia-associated signalling proteins (such as EGFR, GSK3β) and centrosomal proteins (CEP170, which was followed up in detail below, and CEP131) were significantly upregulated by both cilia disassembly stimuli (Fig. 2d).

KSEA, a bioinformatics approach that determines which kinases are differentially-regulated in a sample relative to the control, based on differences in phosphorylation of their substrate groups, expressed as an activity score[63] was then used to determine the kinases active during cilia disassembly. Substrate groups were defined using the OmniPath[71], Edges[72] and PhosphoSitePlus[73] databases. The greatest number of kinases was regulated by 15 min serum treatment (Fig. 2e).

Kinases involved in class I PI3K/AKT and MAPK pathways, as well as PKC and PKA family members were activated by serum and/or LPA in ciliated cells, with multiple cell cycle-associated kinases exhibiting reduced activity (Fig. 2e). Notably, the class I PI3K pathway effector AKT1 was the most consistently upregulated kinase following both treatments and timepoints across the three analyses (Fig. 2e). GDC-0941 had an inhibitory effect on known class I PI3K pathway kinases and further activated multiple MAPKs and PKAs (Fig. 2e). Therefore, these data provide most direct evidence to date that class I PI3Ks are active during cilia disassembly and also suggest the *Pik3ca*[H1047R] embryonic phenotype may, in part, result from increased cilia disassembly.

### PI3Kα and PI3Kβ contribute to cilia disassembly
To test the class I PI3K contribution to cilia disassembly, we performed serum-induced cilia disassembly assays in hTERT-RPE1 cells in the presence or absence of the pan-class I PI3K inhibitor GDC-0941 or the isoform-selective inhibitors BYL719 (targeting PI3Kα), TGX-221 (targeting PI3Kβ), parsaclisib (targeting PI3Kδ) or IPI-549 (targeting PI3Kγ) (Fig. 3a). GDC-0941, BYL719 and TGX-221 partially impaired cilia disassembly compared to DMSO, while parsaclisib and IPI-549 had no effect (Fig. 3b). High expression of PI3Kδ and PI3Kγ is restricted to leukocytes[31], consistent with our finding that these PI3Ks are not relevant in hTERT-RPE1 retinal epithelial cells. Control experiments confirmed that GDC-0941, BYL719 and TGX-221 reduced PI3K/AKT signalling (Fig. 3c). We also confirmed that the inhibitors are active by single-cell analysis of ARL13B-positive ciliated cells in which the serum-induced increase of the mean fluorescence intensity (MFI) of the PI3K/

mTORC1 effector pS6RP(S240/S244) was reduced by BYL719 and GDC-0941 (Fig. 3d).

In the context of a second cilia disassembly inducer, LPA, BYL719, BYL719 + TGX-221 and more effectively GDC-0941 also partially inhibited cilia disassembly, with TGX-221 treatment showing a trend for inhibition (Fig. 3c, e). GSK2636771, a structurally unrelated PI3Kβ inhibitor, partially inhibited serum- and LPA-induced cilia disassembly (Fig. 3f). Taken together, these data indicate that both PI3Kα and PI3Kβ contribute to cilia disassembly.

Primary cilia dynamics were also examined in *Pik3ca*[-/-] mouse embryonic fibroblasts (MEFs)[74]. Following 48 h serum starvation, the percentage of cells that assembled cilia was unchanged in *Pik3ca*[-/-] MEFs compared to wild-type cells (Fig. 3g), however, 24 h or 48 h serum stimulation induced cilia disassembly in *Pik3ca*[+/+] MEFs, an effect which was significantly abrogated upon *Pik3ca* deletion (Fig. 3g).

### PI3Kβ produces the basal PIP₃ pool at the cilia transition zone in hTERT-RPE1 cells, while both PI3Kα and PI3Kβ contribute to stimulus-induced ciliary PIP₃
PIP₃ is present in serum-starved conditions at the cilia transition zone, with its levels increasing in this location upon stimulation with growth factors[23] known to induce cilia disassembly[18,33,34]. Therefore, we next investigated whether PI3Kα and/or PI3Kβ contribute to basal and stimulated ciliary PIP₃.

Ciliated hTERT-RPE1 cells were treated with BYL719, TGX-221 or GDC-0941 for 1 h, fixed and immunostained with well-characterised PIP₃-specific antibodies[28,30] (Fig. 4a). Both TGX-221 and GDC-0941 reduced the basal cilia PIP₃ MFI compared to DMSO control, with BYL719 having no effect (Fig. 4a, Supplementary Fig. 3), suggesting that PI3Kβ contributes to basal ciliary PIP₃.

PI(3,4)P₂ is produced from PIP₃ by inositol polyphosphate 5-phosphatases, and also localises to the transition zone[23]. Similar to PIP₃, the levels of cilia transition zone PI(3,4)P₂, as detected using specific antibodies[23], were reduced in starved cells by TGX-221 and GDC-0941 but not by BYL719 (Fig. 4b), supporting the contention that PI3Kβ is the key active class I PI3K isoform in unstimulated hTERT-RPE1 cells.

Given that PI3Kα did not contribute to ciliary PIP₃ in basal conditions, we next asked whether this kinase plays any role in the ciliary PIP₃ pool by directly activating this kinase using genetic or pharmacological tools. Serum-starved *PIK3CA*[H1047R] cells exhibited an increased transition zone PIP₃ MFI compared to vector control cells (Fig. 4c). Similarly, the allosteric PI3Kα-specific small molecule activator 1938[75] increased the ciliary PIP₃ MFI in starved MEFs and hTERT-RPE1 cells relative to DMSO control (Fig. 4d, e). These data indicate that PI3Kα can produce ciliary PIP₃ under stimulated conditions.

We next assessed whether PI3Kα and/or PI3Kβ contribute to the increase in ciliary PIP₃ levels observed following growth factor stimulation. Ciliated MEFs were treated with EGF for 2 h and the ciliary PIP₃ MFI examined. EGF stimulation increased the PIP₃ levels in *Pik3ca*[+/+] MEFs but not in *Pik3ca*[-/-] MEFs (Fig. 4f). Furthermore, 1 h pretreatment with a PI3Kα inhibitor (BYL719) or a PI3Kβ inhibitor (TGX-221) also

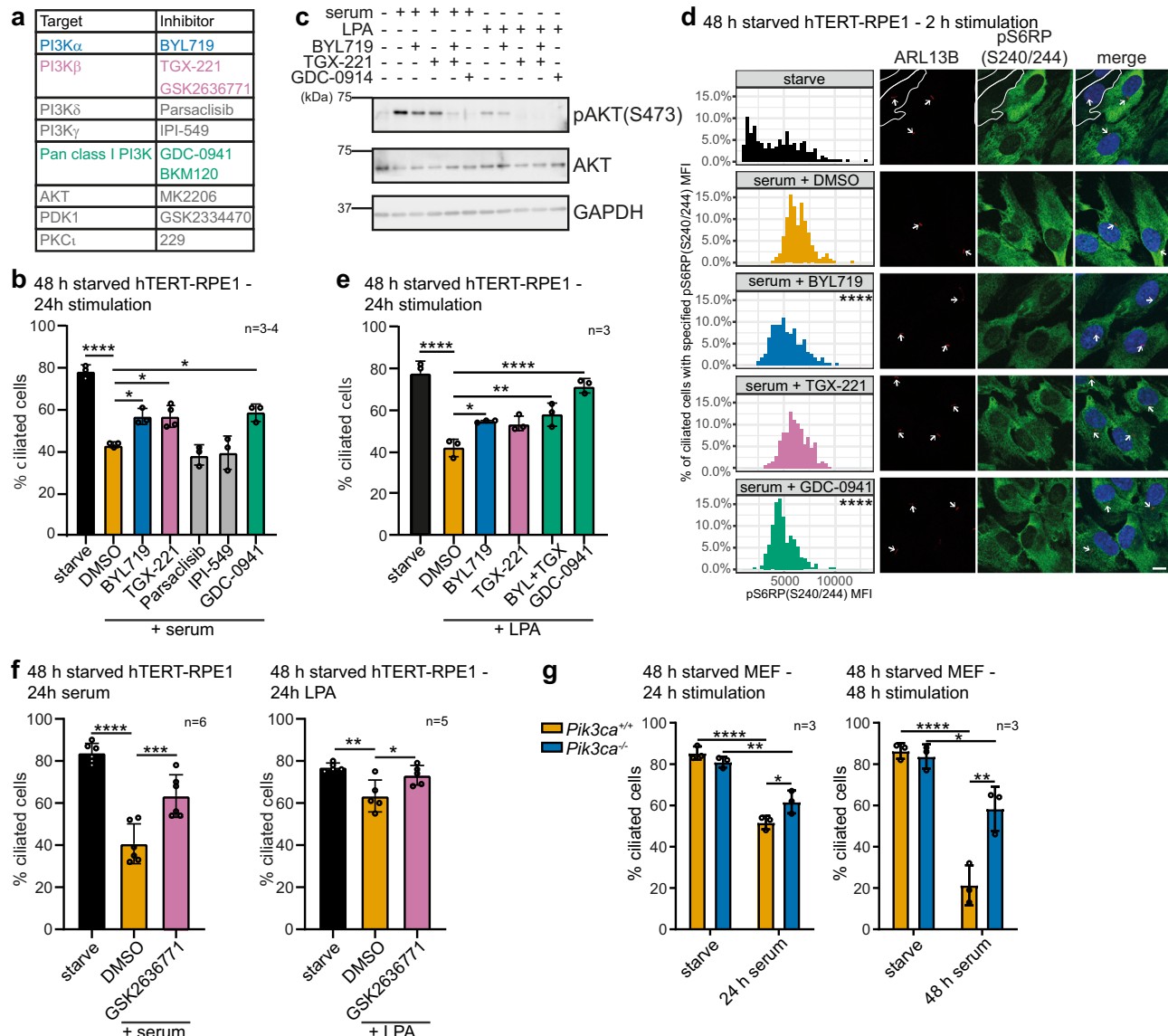

**Fig. 3 | Inhibition of PI3Kα or PI3Kβ impairs cilia disassembly in hTERT-RPE1 cells. a** Inhibitors used. **b** Cells were serum-starved for 48 h, pre-treated with inhibitors or DMSO for 1 h and stimulated with serum in the presence of inhibitors for 24 h. % ciliated cells was scored, bars: mean ± SD, 100 cells/condition for $n = 3,4$ independent experiments, $*p < 0.05$, $****p < 0.0001$ (one-way ANOVA, $p = 3.3 \times 10^{-7}$). **c** Cells were serum-starved for 48 h, pre-treated with inhibitors or DMSO for 1 h and then stimulated with LPA or serum in the presence of inhibitors for 2 h. Lysates were immunoblotted with pAKT(S473), AKT and GAPDH antibodies, representative of $n = 3$ independent experiments. **d** Cells were serum-starved for 48 h, pre-treated with inhibitors or DMSO for 1 h and then stimulated with serum in the presence of inhibitors for 2 h. Cells were stained with ARL13B and pS6RP(S240/244) antibodies and DAPI and imaged by confocal microscopy, arrows: cilia, bar: 10 μm, cells with low pS6RP(S240/244) are indicated by a white outline. pS6RP(S240/244) MFI was measured in ciliated cells and presented as a histogram. $n = 223–254$ cells/condition from 3 independent experiments $****p < 0.0001$ (Kruskal-Wallis test,

$p = 2.59 \times 10^{-53}$). **e** Cells were serum-starved for 48 h, pre-treated with inhibitors or DMSO for 1 h and then stimulated with LPA in the presence of inhibitors for 24 h. % ciliated cells was scored, bars indicate mean ± SD, 100 cells/condition for $n = 3$ independent experiment, $*p < 0.05$, $**p < 0.01$, $****p < 0.0001$ (one-way ANOVA, $p = 5.032 \times 10^{-6}$). **f** Cells were serum-starved for 48 h, pre-treated with GSK2636771 or DMSO for 1 h and then stimulated with serum (left) or LPA (right) in the presence of inhibitors for 24 h. % ciliated cells was scored, bars: mean ± SD, 100 cells scored/condition for $n = 6$ (left), $n = 5$ (right) independent experiments, $*p < 0.05$, $**p < 0.01$, $***p < 0.001$, $****p < 0.0001$ (one-way ANOVA, left $1.06 \times 10^{-6}$, right $p = 0.0043$). **g** $Pik3ca^{+/+}/Pik3ca^{-/-}$ MEFs were serum-starved for 48 h, ± 24 or 48 h serum stimulation. % ciliated cells was scored, bars: mean ± SD, 100 cells/condition for $n = 3$ independent experiments, $*p < 0.05$, $**p < 0.01$, $****p < 0.0001$ (two-way ANOVA, left interaction $p = 0.0155$, row $p = 2.072 \times 10^{-6}$, column $p = 0.2335$, right interaction $p = 0.00267$, row $p = 1.022 \times 10^{-5}$, column p = 0.00598). Source data are provided as a Source Data file.

abrogated the EGF-induced increase in ciliary PIP$_3$ MFI in wild-type MEFs (Fig. 4g, h), indicating that PI3Kα and PI3Kβ contribute to the stimulus-induced increase in PIP$_3$ at the cilia transition zone. However, we noted that in MEFs the basal ciliary PIP$_3$ levels were more difficult to detect than in hTERT-RPE1 cells, with TGX-221 having little effect in unstimulated MEFs (Fig. 4h).

Taken together, our data show that the basal transition zone PIP$_3$ pool in hTERT-RPE1 cells is produced by PI3Kβ, whereas both PI3Kα

and PI3Kβ contribute to the increase in PIP$_3$ levels at cilia following agonist stimulation.

## PI3Kα initiates cilia disassembly

Primary cilia disassembly is defined by two phases, namely the initiation of cilia disassembly and an ongoing suppression of cilia re-assembly[21]. Distinct proteins have been linked to each of these phases. We next investigated which step in cilia disassembly class I PI3Ks may

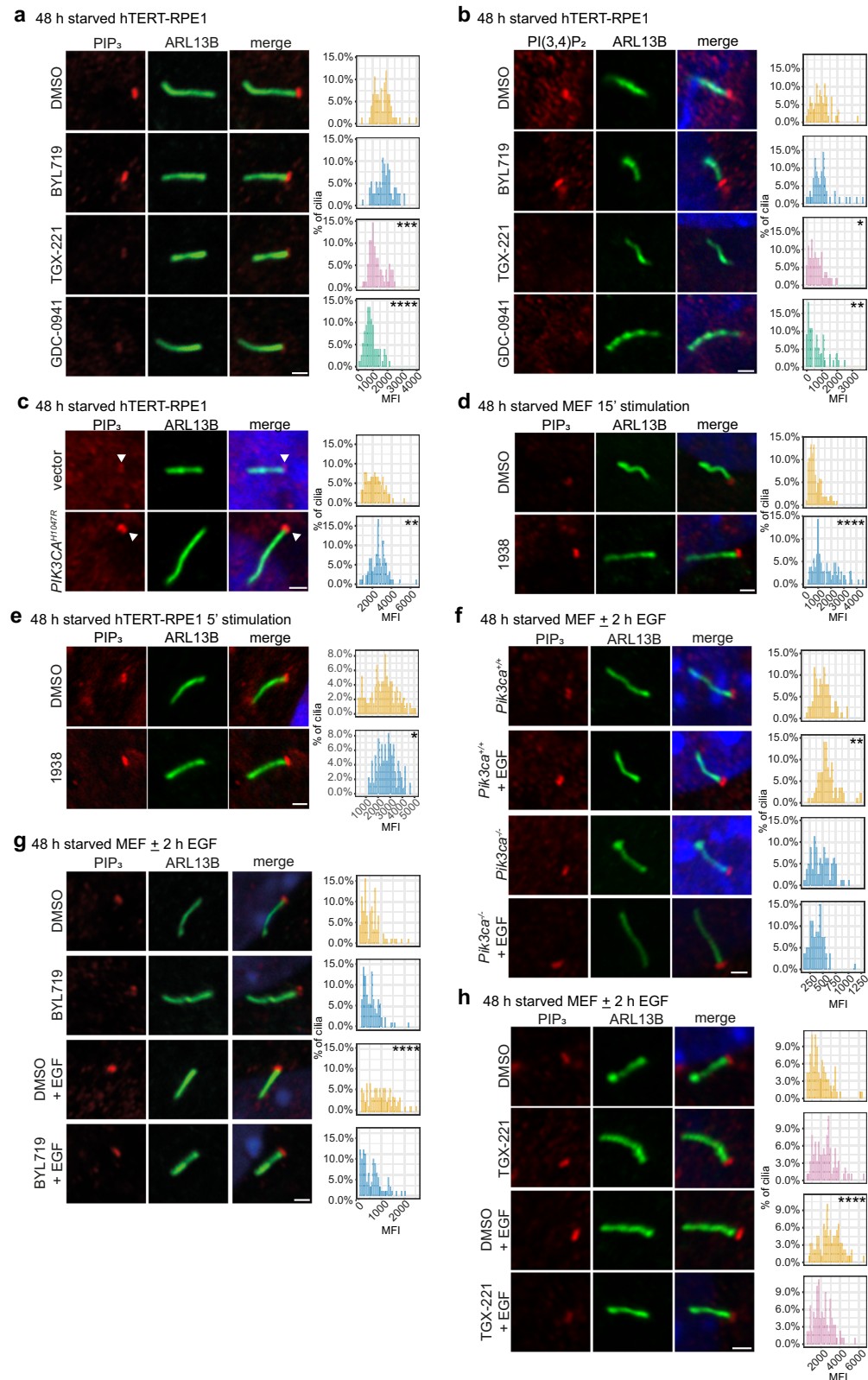

act. Additionally, our loss-of-function studies described above show that PI3Kα and PI3Kβ are necessary for cilia disassembly but not whether their activation is sufficient for the process. For the following studies, we focused on PI3Kα given the cilia-dependent phenotypes observed in *Pik3ca*[H1047R] embryos and its significant role in human disease[76,77].

Firstly, we showed that the reduction of *PIK3CA*[H1047R]-expressing hTERT-RPE1 cells that form cilia upon serum starvation (Supplementary Fig. 1e) was rescued by BYL719 (Fig. 5a). Doxycycline-induced *PIK3CA*[H1047R] expression in serum-starved MCF10A cells (an immortalised but not transformed human breast cell line) also reduced the percentage of ciliated cells (Supplementary Fig. 4a–d). In complete

**Fig. 4 | Basal ciliary PIP$_3$ is produced by PI3Kβ in hTERT-RPE1 cells, with both PI3Kα and PI3Kβ contributing to the stimulus-induced ciliary PIP$_3$ increase.** hTERT-RPE1 cells were serum-starved for 48 h and treated with BYL719, TGX-221, GDC-0941 or DMSO for 1 h. Cells were, stained with ARL13B and **(a)** PIP$_3$ or **(b)** PI(3,4)P$_2$ antibodies and DAPI and imaged by confocal microscopy, bar: 1 μm. PI MFI was measured, **(a)** $n > 75$ or **(b)** $n = 54–55$ cells/condition from 3 independent experiments *$p < 0.05$, **$p < 0.01$, ***$p < 0.001$, ****$p < 0.0001$ relative to DMSO control (Kruskal-Wallis test, **(a)** $p = 8.440 \times 10^{-21}$, **(b)** $p = 1.30 \times 10^{-4}$). hTERT-RPE1(Vector/*PIK3CA$^{H1047R}$*) **(c)**, MEFs **(d)** or hTERT-RPE1 **(e)** cells were serum-starved for 48 h **(c)** with doxycycline or **(d, e)** stimulated with 1938 or DMSO for 15 or 5 min. Cells were stained with ARL13B and PIP$_3$ antibodies and DAPI and imaged by confocal microscopy, bar: 1 μm, arrowhead: ciliary PIP$_3$. PIP$_3$ MFI was measured, **(c)** $n = 90$, **(d)** $n = 105$ or **(e)** $n = 131–133$ cells/condition from 3 independent experiments *$p < 0.05$, **$p < 0.01$, ****$p < 0.0001$ (two-sided Kolmogorov-Smirnov test **(c)** $p = 0.0042$, **(d)** $p = 5.907 \times 10^{-10}$ **(e)** $p = 0.0437$). **f** MEFs(*Pik3ca$^{+/+}$/Pik3ca$^{-/-}$*) were serum-starved for 48 h and EGF stimulated for

2 h. Cells were stained with ARL13B and PIP$_3$ antibodies and DAPI and imaged by confocal microscopy, bar: 1 μm. PIP$_3$ MFI was measured, $n = 80–85$ cells/condition from 3 independent experiments **$p < 0.01$ relative to untreated control cells (Kruskal-Wallis test, $p = 8.836 \times 10^{-13}$). MEFs were serum-starved for 48 h, treated with **(g)** BYL719, **(h)** TGX-221 or DMSO for 1 h and stimulated ± EGF for 2 h in the presence or absence of inhibitors. Cells were, stained with ARL13B and PIP$_3$ antibodies and DAPI and imaged by confocal microscopy, bar: 1 μm. PIP$_3$ MFI was measured, $n = 90$ cells/condition from 3 independent experiments ****$p < 0.0001$ relative to DMSO control (Kruskal-Wallis test, **(g)** $p = 9.870 \times 10^{-9}$, **(h)** $p = 5.869 \times 10^{-9}$). **a–h** For all PI imaging experiments using different treatments, the laser intensity, gain and brightness were adjusted independently to optimise the dynamic range of the experiment and applied to all conditions within the experiment. To measure the ciliary PI MFI, for each cilium, a box of standardised size was placed at the base of the ARL13B demarked axoneme centred around the highest intensity PI pixel and MFI within the box measured and presented as a histogram.

---

media, hTERT-RPE1 cells exhibit primary cilia, but at a lower frequency than in starved conditions. *PIK3CA$^{H1047R}$* expression in these cells reduced the percentage of spontaneously ciliated cells in complete media compared to vector control (Fig. 5b). These cilia phenotypes in *PIK3CA$^{H1047R}$* cells could be interpreted as an increase in cilia disassembly or inhibition of ciliogenesis. However, combined with our PI3K inhibitor and knockout studies described above, the most likely explanation is continuous activation of cilia disassembly. Nevertheless, to strengthen the contention that PI3Kα activation perturbs cilia disassembly rather than assembly, we examined cilia length over a time course of serum-starvation in *PIK3CA$^{H1047R}$* and vector control hTERT-RPE1 cells. A defect in cilia assembly manifests as a reduction in cilia length, evident within hours of serum withdrawal[78]. Consistent with PI3Kα playing a predominant role in cilia disassembly, *PIK3CA$^{H1047R}$* hTERT-RPE1 cells exhibited no change in cilia length over 48 h of serum starvation (Fig. 5c).

To assess whether acute pharmacological activation of PI3Kα in ciliated cells is sufficient to induce cilia disassembly, similar to cilia disassembly induced by serum, wild-type MEFs were serum-starved for 48 h followed by stimulation with the 1938 PI3Kα activator for 24 h, resulting in a subtle but significant induction of cilia disassembly (Fig. 5d). In hTERT-RPE1 cells, serum treatment for 2, 4, 8 or 24 h induced cilia disassembly over the time course (Fig. 5e). In contrast, 1938 induced cilia disassembly up to 8 h, but surprisingly by 24 h cilia started to re-form (Fig. 5e). Notably, 1938 and serum both transiently activated PI3K/AKT signalling to a similar extent at these timepoints (Fig. 5e), suggesting PI3Kα activation initiates axoneme resorption but activation of other pathways by serum is required to sustain the second phase of cilia disassembly (i.e. the inhibition of ongoing re-assembly).

PI3Kα is known to promote proliferation and cilia are linked to the cell cycle, with the basal body acting as the mitotic spindle. This raises the question of whether the cilia phenotypes in *PIK3CA$^{H1047R}$* hTERT-RPE1 cells are due to a direct effect of PI3Kα on cilia biology rather than a secondary consequence of perturbed cell cycle kinetics. There are numerous published examples of cilia defects which are not associated with changes in the cell cycle[12,79–82]. Consistently, in cells expressing *PIK3CA$^{H1047R}$* or with *Pik3ca* deletion under the culture conditions in which we observed cilia phenotypes, we did not observe obvious changes in cell proliferation, except for an ~5% increase in EdU incorporation in *PIK3CA$^{H1047R}$* hTERT-RPE1 cells relative to controls (Supplementary Fig. 5a-e) which is unlikely to be the sole cause of the ~30% reduction in ciliated hTERT-RPE1 cells (Supplementary Fig. 1e).

### PI3Kα inhibition partially rescues cilia in hyperplastic and cancer cells

We next asked whether PI3K inhibition restores cilia in benign hyperplastic and cancer cell line models known to be capable of forming cilia at a low level under defined experimental conditions.

We first tested the human BPH-1 benign prostatic hyperplasia cell line and observed the pan-PI3K inhibitor BKM120 increased the proportion of ciliated cells (Fig. 5f), a phenomenon that occurred in the absence of accumulation of cells in G0/G1, the cell cycle phase where cilia assembly is promoted (Fig. 5g. Supplementary Fig. 5f).

The *KRAS*-mutant human A549 lung cancer cell line exhibits hyperactivation of PI3K signalling[83]. Approximately 30% of these cells exhibit cilia in starved conditions, with inhibition of PI3Kα or class I PI3Ks (using BYL719 or GDC-0941, respectively) increasing the proportion of ciliated cells, in contrast to the PI3Kβ inhibitor TGX-221 which had no effect (Fig. 5h). In line with these data, *PIK3CA* deletion by CRISPR-Cas9 (Supplementary Fig. 5g) also increased the proportion of ciliated A549 cells (Fig. 5i). Of note, the proportion of ciliated A549 cells remained relatively low upon PI3K inhibition or deletion, with a maximum of ~45% ciliated, indicating that hyperactive pathways in these cancer cells other than PI3K are likely to also contribute to the loss of cilia. Our observations in A549 cells are in line with data on murine pancreatic cancer cells where *Kras* mutations reduce the fraction of ciliated cells, which is rescued by LY294002, an inhibitor which targets PI3K among many other kinases[11,36,37].

### PI3Kα induces CEP170 phosphorylation and activates PKCι

To further examine the signalling network activated by PI3Kα/PIP$_3$ during cilia disassembly, we performed a second phosphoproteomic experiment in ciliated hTERT-RPE1 cells using 15 min or 4 h treatment with 1938 as a more specific PI3Kα stimulus than serum or LPA. We also included treatment with insulin, a canonical PI3Kα stimulus, and BYL719 as controls (Supplementary Fig. 6a, b). Phosphosites exhibiting a > 2-fold change relative to DMSO and adjusted *p*-value < 0.05 were defined as significantly regulated. We quantified 8544 phosphosites (Supplementary Data 3) from 2648 proteins including 7266 pSer, 1179 pThr and 99 pTyr residues (Supplementary Fig. 6a) and phosphorylation of several known PI3K pathway targets, validating the approach (Fig. 6a, Supplementary Fig. 6d).

Notably, among the novel PI3Kα-regulated phosphosites, this experiment, similar to the serum/LPA phosphoproteomic experiment above (Fig. 2d), revealed differential phosphorylation of CEP170 (Fig. 6a), which we prioritised as a candidate PI3Kα effector in cilia disassembly. CEP170 is a centrosomal protein localised to the proximal end of the centrosome and the subdistal appendages, known to induce cilia disassembly by recruitment of the microtubule depolymerising kinesin KIF2A[20,84,85]. Similar to *Pik3ca* deletion (Fig. 3g), *CEP170* and *KIF2A* knockdown induce impaired or delayed cilia disassembly[20,85].

Phosphorylation of CEP170(S466) was observed upon 15 min stimulation with 1938, and to a lesser extent with insulin, and in both cases was ablated by BYL719 (Fig. 6a, b). pCEP170(S466) was also observed in our phosphoproteomic analysis of serum- and LPA-stimulated ciliated hTERT-RPE1 cells (Fig. 2d) and in MEFs treated with 1938 (S463

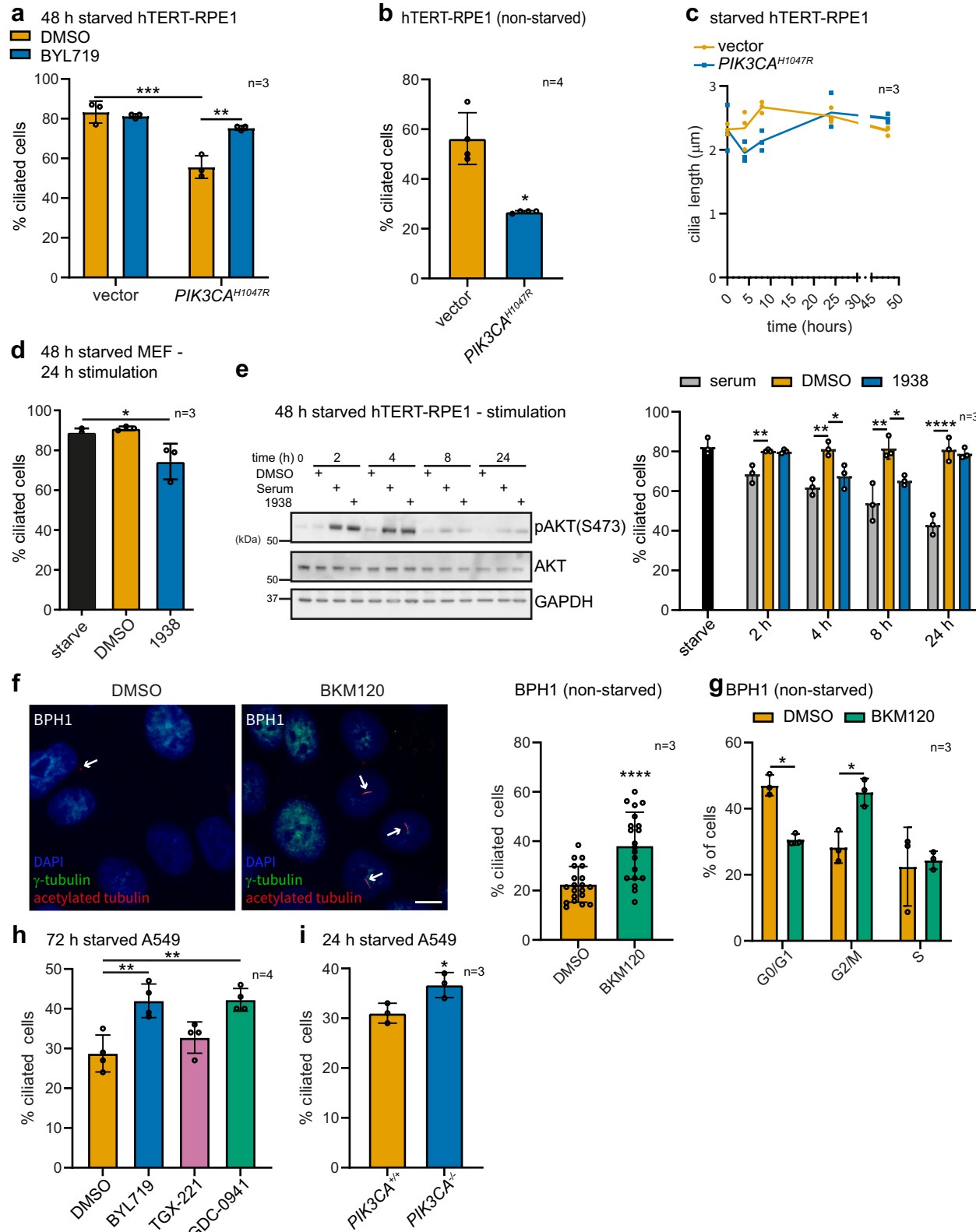

conserved in murine CEP170 Fig. 6c; Fig. 4a of Ref. 75.), which was neutralised by GDC-0941 or BYL719, respectively. Whereas, the CEP170(S466) phosphosite has been reported in several phosphoproteomic studies[86–98] and catalogued in PhosphoSitePlus, to the best of our knowledge no functional characterisation has been performed.

To assess the functional impact of the S466 phosphosite in CEP170, we expressed wild-type (HA-CEP170) and a phosphomimetic

(HA-CEP170(S466D)) mutant of CEP170 in HEK293 cells. Although the proportion of serum-starved HEK293 cells that form cilia is relatively low, this cell line has been used previously to study primary cilia[99] and was chosen here given its relative ease of transfection and the fact that we were unable to express HA-CEP170 plasmids in hTERT-RPE1 cells, likely due to the large plasmid size (4755 nucleotides). Wild-type and mutant CEP170 predominantly localised to the centrosome (Fig. 6d).

**Fig. 5 | PI3Kα activation initiates cilia disassembly. a** hTERT-RPE1(Vector/ *PIK3CA*^*H1047R*) cells were serum-starved for 48 h with doxycycline and treated ± BYL719 for the final 24 h. % ciliated cells was scored, bars indicate mean ± SD, 100 cells scored/condition for *n* = 3 independent experiments, **$p < 0.01$, ***$p < 0.001$ (two-way ANOVA, interaction $p = 0.0017$, row $p = 9.117 \times 10^{-5}$, column $p = 0.0053$). **b** hTERT-RPE1(Vector/*PIK3CA*^*H1047R*) cells were treated with doxycycline for 48 h in complete media. % of ciliated cells was scored, bars indicate mean ± SD, 100 cells scored/condition for *n* = 4 independent experiments, *$p < 0.05$ (two-sided Student's *t*-test, $p = 0.0108$). **c** hTERT-RPE1(Vector/*PIK3CA*^*H1047R*) cells were treated with doxycycline for 24 h in complete media then serum starved for up to 48 h. Cilia length was measured, line: mean, cilia from ≥13 cells/condition for *n* = 3 independent experiments. **d** MEFs were serum-starved for 48 h and stimulated with 1938 or DMSO for 24 h. % ciliated cells was scored, bars indicate mean ± SD, 100 cells scored/condition for *n* = 3 independent experiments, *$p < 0.05$ (one-way ANOVA, $p = 0.0167$). **e** hTERT-RPE1 cells were serum-starved for 48 h and stimulated with serum, 1938 or DMSO for up to 24 h. Left, cells immunoblotted with pAKT(S473), AKT and GAPDH antibodies, blots representative of *n* = 3 independent experiments. Right, % of ciliated cells was scored, bars: mean ± SD, 100 cells/condition for

*n* = 3 independent experiments, *$p < 0.05$, **$p < 0.01$, ****$p < 0.0001$ (one-way ANOVA, post hock test relative to DMSO treatment at each time point, 2 h $p = 0.0026$, 4 h $p = 0.0013$, 8 h $p = 0.0011$, 24 h $p = 1.533 \times 10^{-5}$). **f** BPH1 cells were treated with BKM120 for 8 h, stained with acetylated tubulin and γ tubulin antibodies and DAPI, bar: 10 μm, arrows: cilia. Right, % ciliated cells was scored. Bars represent mean ± SD, ≥ 237 cells from 20 micrographs pooled from 3 independent experiments, ****$p < 0.0001$ (two-sided Student's *t*-test $p = 8.588 \times 10^{-5}$). **g** BPH1 cells were treated with BKM120 for 8 h and cell cycle distribution assessed by flow cytometry of propidium iodide-stained cells. Bars: mean ± SD, *n* = 3 independent experiments, *$p < 0.05$, (two-way ANOVA, interaction $p = 0.0013$, row $p = 0.0012$, column $p = 0.7914$). **h** A549 cells were serum-starved for 72 h ± BYL719, TGX-221, GDC-0941 or DMSO and % ciliated cells scored, bars: mean ± SD, 100 cells/ condition for *n* = 4 independent experiments, **$p < 0.01$ (one-way ANOVA, $p = 0.0007$). **i** A549(*PIK3CA*^+/+/*PIK3CA*^-/-) cells were serum-starved for 24 h and % ciliated cells scored, bars represent mean ± SD, 100 cells scored/condition for *n* = 3 independent experiments, *$p < 0.05$ (two-sided Student's *t*-test $p = 0.0379$). Source data are provided as a Source Data file.

Interestingly, expression of HA-CEP170(S466D) reduced the proportion of ciliated transfected cells compared to wild-type CEP170 (Fig. 6e), indicating that phosphorylation of S466 regulates the ability of CEP170 to control cilia disassembly.

As pCEP170(S466) has not been previously characterised as a target of PI3K signalling, we investigated which PI3K-dependent kinase may be responsible for its phosphorylation, using KSEA with the OmniPath[71], Edges[72] and PhosphoSitePlus[73] databases. As expected, PI3Kα and its downstream effectors including AKT1, AKT2, AKT3, mTOR and PRAS40 were activated upon 1938 treatment (Fig. 6f, g, Supplementary Fig. 6e, f). Notably, PKCι and A-RAF were the only kinases detected in all three analyses to be activated by 1938 (Fig. 6f, g, Supplementary Fig. 6e, f). PKCι activation was also observed via KSEA using the Edges database in serum-stimulated hTERT-RPE1 cells (Fig. 2e).

Kinase prediction analysis of the CEP170(S466) substrate motif using the NetPhos 3.1 Server[100] [https://services.healthtech.dtu.dk/ services/NetPhos-3.1/] identified the PKC family as the top candidate kinases, of which PKCι, a member of the atypical PKC family, was the only PKC isoform activated by 1938 in our KSEA (Fig. 6g). Furthermore, PKCι was the 6th ranked putative kinase with a site percentile of 98% for this phosphosite in the Johnson et. al. 2023[101] kinase-substrate atlas [https://kinase-library.phosphosite.org/] (accessed 31/05/2023) and the only candidate in the top 20 ranks that we observed to be consistently activated by 1938. Indeed, an in vitro kinase assay using purified recombinant PKCι kinase domain (phosphorylated at activation loop and turn motif priming sites) and a CEP170(S466) peptide revealed phosphorylation of this site by PKCι (Fig. 6h).

PKCι activity is known to be regulated by PI3K/ PDK1 signalling[102–105]. Interestingly, there is some indication for an involvement of PKCι in cilia disassembly, whereby it localises to the cilia base and upon overexpression reduces ciliated cells in some instances, yet no mechanistic investigation has been reported[106–108]. Our analysis validates these findings and clearly positions PKCι in the cilia disassembly signalling network. We propose a working model in which CEP170 is phosphorylated at S466 by PKCι downstream of PDK1, contributing to PI3Kα-dependent cilia disassembly.

KSEA also identified AKT activation in serum-, LPA- and 1938-stimulated ciliated cells (Fig. 2e, Fig. 6f, g, Supplementary Fig. 6e, f). Given the AKT substrate GSK3β is an established regulator of ciliary microtubule stability, we previously proposed increased PIP₃ levels in *Inpp5e* null cells may promote cilia disassembly via AKT/GSK3β signalling[23]. It is likely this mechanism also contributes in the context of PI3Kα activation. Interestingly, we did not observe increased activation of AURKA, one of the most well-established cilia

disassembly kinases, in 1938 stimulated cells (Fig. 6f, g, Supplementary Fig. 6f).

## PDK1, PKCι, CEP170 and KIF2A contribute to PI3Kα-dependent cilia disassembly

Using rescue experiments, we next investigated the functional impact on cilia disassembly of the above-identified kinase signalling pathways downstream of PI3Kα. hTERT-RPE1 cells inducibly expressing vector-control or *PIK3CA*^*H1047R* were serum-starved and treated with doxycycline ± kinase inhibitors or siRNA to inhibit or knockdown putative pathway components. We first tested inhibitors of PDK1 (GSK2334470), AKT (MK2206) or PKCι (229) and observed that they did not affect the percentage of ciliated vector-control cells but rescued the percentage of *PIK3CA*^*H1047R*-expressing cells that displayed cilia (Fig. 7a–c). siRNA knockdown of *PRKCI* also partially rescued the percentage of ciliated *PIK3CA*^*H1047R* cells compared to non-targeting siRNA control (Fig. 7d, Supplementary Fig. 7a). Similarly, knockdown of *CEP170* or its effector *KIF2A* rescued the percentage of ciliated *PIK3CA*^*H1047R* cells compared to non-targeting control, with no effect on vector control cells (Fig. 7e, f, Supplementary Fig. 7b–d).

To compare the functional contribution of each kinase to cilia disassembly more directly, *PIK3CA*^*H1047R* hTERT-RPE1 cells were treated with inhibitors to each kinase in the putative pathway in a single experiment. In agreement with our data above and in line with a phenotype driven by activation of PI3Kα, all inhibitors except for the PI3Kβ inhibitor TGX-221, almost completely rescued the percentage of ciliated cells to vector control levels, with little difference in efficacy between inhibitors (Fig. 7g). In the context of serum-induced cilia disassembly in parental hTERT-RPE1 cells, inhibitors of PI3Kα, PI3Kβ, pan-class I PI3K, PDK1, AKT or PKCι all induced a partial rescue of cilia disassembly, with a similar efficacy observed amongst inhibitors (Fig. 7h, Supplementary Table 3). The partial rescue following serum-induced cilia disassembly (Fig. 7h) compared to the almost complete rescue in *PIK3CA*^*H1047R* cells (Fig. 7g) indicates that PI3K and downstream signalling is only one, but a physiologically critical, component of the serum-induced cilia disassembly network.

## Discussion

PI membrane lipids define membrane identity and orchestrate most aspects of cell physiology[25]. This includes the organisation of primary cilium membrane subdomains which are characterised by the presence of specific PIs[23,26–28,30]. However, the kinases responsible for agonist-induced PI generation at cilia, their effector function in cilia biology and the possible associated consequences for human disease remain unclear.

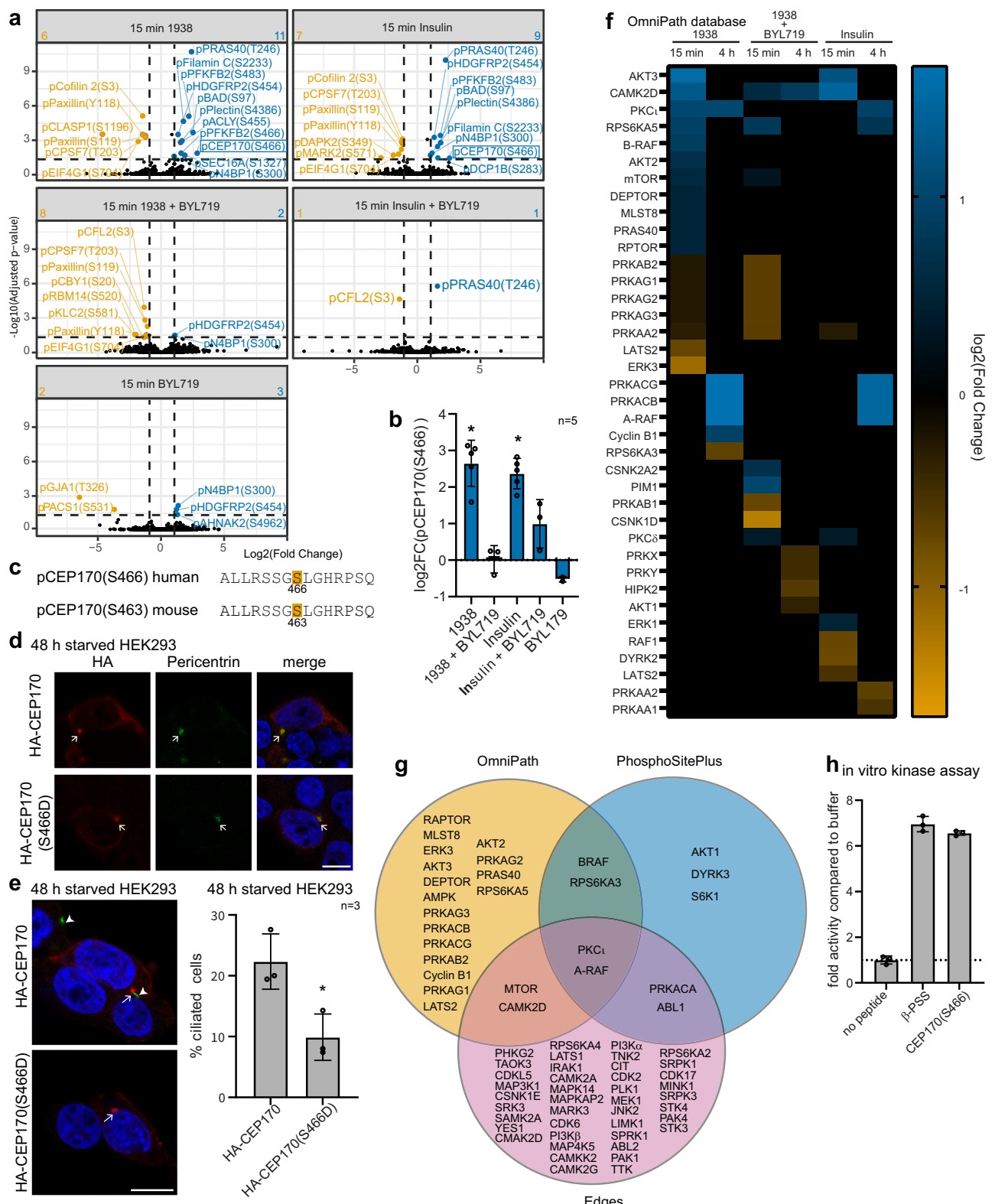

In this study, we uncovered key roles for the ubiquitously-expressed PI3Kα and PI3Kβ class I PI3K isoforms in cilia biology. Ubiquitously-expressed activating *PIK3CA* mutations have not been identified in humans, likely due to early embryonic lethality. However, mosaic activating *PIK3CA* mutations are observed in PROS, characterised by early onset segmental overgrowth and vascular malformations[109]. While these phenotypes appear to be related to classical PI3K functions such as in vascular development and growth,

some affected individuals also exhibit characteristic ciliopathy phenotypes such as polydactyly and renal cysts[109–112]. Polydactyly is caused by dysregulation of GLI3 processing in the primary cilium[52]. The function of PI3Kα in cilia disassembly identified here, the cilia-related phenotypes of *Pik3ca*$^{H1047R}$ embryos and the presence of polydactyly and renal cysts in PROS provide supporting evidence to speculate PROS may be classified as a DCC, where the mutant protein exhibits both ciliary and non-ciliary functions and some, but not all, phenotypic

**Fig. 6 | PI3Kα signalling induces CEP170 phosphorylation and PKCι activation.** **a, b** Phosphoproteomic analysis of hTERT-RPE1 cells serum-starved for 24 h and stimulated with 1938 or insulin ± BYL719 for 15 min (*p*-value calculated using the group comparison function within MSstats and adjusted using the Benjamini-Hochberg procedure, *n* = 5 independent experiments). **a** Volcano plot of phosphosites differentially regulated by 1938 or insulin ± BYL719 relative to DMSO-treated cells. Numbers in the top corners indicate numbers of phosphosites significantly up- or down-regulated relative to DMSO **b** Log2(FC) of pCEP170(S466), bars: mean ± SD, *\*p* < 0.05 (1938 vs DMSO *p* = 0.01438, insulin vs DMSO *p* = 0.04240). **c** Sequence alignment of human and mouse CEP170(S466/S463). HEK293 cells were transfected with HA-CEP170 or HA-CEP170(S466D) plasmids, serum-starved for 48 h, fixed and stained with HA and (**d**) pericentrin or (**e**) ARL13B antibodies and DAPI, bar: 10 μm, arrows indicate centrosomes, arrowheads: cilia axonemes. **d** Representative of *n* = 3 independent experiments. **e** The % of ciliated transfected cells was scored, bars: mean ± SD, ≥21 transfected cells scored/condition for *n* = 3 independent experiments, *\*p* < 0.05 (two-sided Student's *t*-test,

*p* = 0.0221). **f, g** KSEA (OmniPath) of kinases for which the substrate groups are differentially regulated by 15 min or 4 h 1938 or Insulin stimulation ± BYL719 in 24 h serum-starved hTERT-RPE1 cells relative to DMSO. As few phosphosites were altered by 1938 or insulin treatment, kinases for which the raw *p*-values relative to DMSO (using the Kolmogorov–Smirnov test) relative to DMSO control were *p* < 0.05 were considered significantly regulated. **f** Heatmap. **g** Venn diagram showing overlap of kinases differentially regulated in KSEA (OmniPath vs Edges vs PhosphoSitePlus database). **h** In vitro kinase assay for purified recombinant PKCι kinase domain (phosphorylated at activation loop and turn motif priming sites) with β-PSS positive control (pseudosubstrate sequence of PKCβ, with alanine mutated to phosphoacceptor serine, containing the PKCι recognition motif; Phenylalanine at -5 and Arginine at -3 with respect to the phospho-acceptor) and CEP170(S466) peptides, bars represent mean ± SD, *n* = 3 technical replicates, data representative of 2 independent experiments. Source data are provided as a Source Data file.

features are caused by cilia dysfunction[3]. It is conceivable that previously reported phenotypes in PROS are biased towards more obvious external features, with for example, mild renal cysts possibly missed. Our report may therefore help to further diagnose cilia-related phenotypes in PROS-affected individuals.

The other major disease context in which PI3Kα is frequently mutationally-activated is cancer[31]. Numerous studies have reported cilia loss in multiple solid tumours relative to the non-transformed cells of origin[11–15] and postulated that this event promotes tumorigenesis. Indeed, we show PI3K inhibition in A549 lung cancer cells partially rescues the ability of the population to express cilia. Perhaps cancer may also be classified as a DCC.

In our study, we further dissected the signalling pathways leading to and from the PIP$_3$ lipid (Fig. 7i). We show that in hTERT-RPE1 cells, the PIP$_3$ pool in the cilia transition zone is produced by PI3Kβ under basal, stable cilia conditions. Biochemical evidence for a pool of PI3Kβ-dependent PIP$_3$ has been found in specific cell types under basal, non-acutely stimulated culture conditions[113–115], but its subcellular location has not been defined. It is tempting to speculate that the PIP$_3$ in these studies was at least partially associated with primary cilia. At present, it remains unclear how PI3Kβ is activated and a PIP$_3$ pool maintained at cilia in starved cells, however, given the evidence that PI3Kβ can be activated by integrins[116], we hypothesise that cell adhesion-associated signalling may be involved.

Upon stimulation with cilia disassembly mediators, both PI3Kα and PI3Kβ were found to contribute to the increase in PIP$_3$ levels at the cilia transition zone (Fig. 7i). These class I PI3K isoforms phosphorylate PI(4,5)P$_2$ to produce PIP$_3$, and indeed PI(4,5)P$_2$ localises in close proximity to PIP$_3$ at the inner leaflet of the transition zone membrane, albeit in a slightly different axial plane[28,30]. Class I PI3Ks are distinct from the class II/III PI3Ks that control intracellular vesicular transport, mostly in an agonist-independent manner[31], and produce PI(3)P at the pericentrosomal recycling endosomal compartment near the cilia base[117]. It is interesting to speculate that PI3Kα and PI3Kβ may constitutively or dynamically localise to the transition zone and phosphorylate local PI(4,5)P$_2$ in response to growth factor receptor stimulation to produce PIP$_3$. However, the dearth of antibodies to assess class I PI3K subcellular localisation and loss of kinase activity caused by N-terminal or C-terminal epitope tagging PI3Ks[118,119] precludes such analysis.

Cilia disassembly is complex and involves activation of multiple kinase-effector partners[16–21], but key growth factor-effector kinases, known to be downstream of serum and LPA in non-ciliated cells, such as PI3K/AKT, MAPK, PKC and PKA had not been implicated in cilia disassembly. Our untargeted phosphoproteomic analysis in ciliated long-term starved cells, showed that serum and LPA indeed activate these kinases during cilia disassembly providing new candidates to test in focused cilia disassembly assays, as performed here for PI3K.

However, PI3K and downstream effector inhibitors do not fully inhibit serum-induced cilia disassembly to the levels observed in starved cells, but completely rescue *PIK3CA^{H1047R}*-induced cilia instability. This observation indicates that although the pathway is an important component of the cilia disassembly mechanism which induces cilia-dependent phenotypes when perturbed, it is only a subset of the full cellular cilia disassembly network. Our data support a working model in which cilia disassembly stimulation induces PI3Kα and PI3Kβ to increase transition zone PIP$_3$ levels above a certain threshold, with this lipid pool orchestrating parallel effector pathways.

Firstly using AKT inhibitors in our PI3K-activated models, we confirmed the AKT/GSK3β cilia disassembly pathway that we previously proposed in *Inpp5e*-null medulloblastoma cells[23]. Further studies using untargeted approaches allowed us to identify a signalling pathway through the PDK1 and PKCι protein kinases, the centrosomal CEP170 protein and the microtubule depolymerising kinesin KIF2A, most likely parallel to AKT (Fig. 7i).

PKCι is regulated by PI3K/PIP$_3$/PDK1[102–105] with both PDK1 and PKCι localising in proximity to PIP$_3$ at the cilia base[23,107]. Our model, supported by in vitro kinase assays, proposes that activated PKCι phosphorylates the newly-identified S466 phosphorylation site in CEP170 downstream of serum, LPA and 1938 which is neutralised by GDC-0941 and BYL719. Interestingly, in the cilia-retaining Hedgehog-dependent tumour basal cell carcinoma, PKCι is upregulated by Hedgehog signalling and potentiates the Hedgehog pathway via GLI1 phosphorylation[106]. The initiation of cilia disassembly by PKCι may therefore form a negative-feedback-loop controlling oncogenic Hedgehog signalling in this context.

CEP170 is known to induce cilia disassembly via recruitment of the microtubule depolymerising kinesin KIF2A[20,85], therefore we propose CEP170(S466) phosphorylation modulates the interaction. In support of this model, serum-induced cilia disassembly and oncogenic PI3Kα-induced cilia defects are rescued by inhibitors or siRNA to PDK1, PKCι, CEP170 or KIF2A. Of the two cilia disassembly phases; initiation of axoneme resorption and ongoing suppression of cilia re-assembly, KIF2A mediates axoneme resorption[20]. Consistently our studies using the small molecule PI3Kα activator 1938 indicate that PI3Kα activation also initiates axoneme resorption but does not impact sustained suppression of cilia re-assembly. Given this observation, the more subtle cilia phenotype caused by *PIK3CA^{H1047R}* expression compared to serum stimulation and the only partial rescue of serum- but complete rescue of *PIK3CA^{H1047R}*-induced cilia disassembly by pathway inhibitors, we stress that PI3K signalling is a physiologically-relevant part of a broader complex cilia disassembly network. An outstanding question in the field is how PIP$_3$ stimulates PDK1 to phosphorylate the subset of its substrates which themselves do not bind PIP$_3$[120] and how PKCι is regulated by phosphorylation, but these questions are beyond the scope of this study.

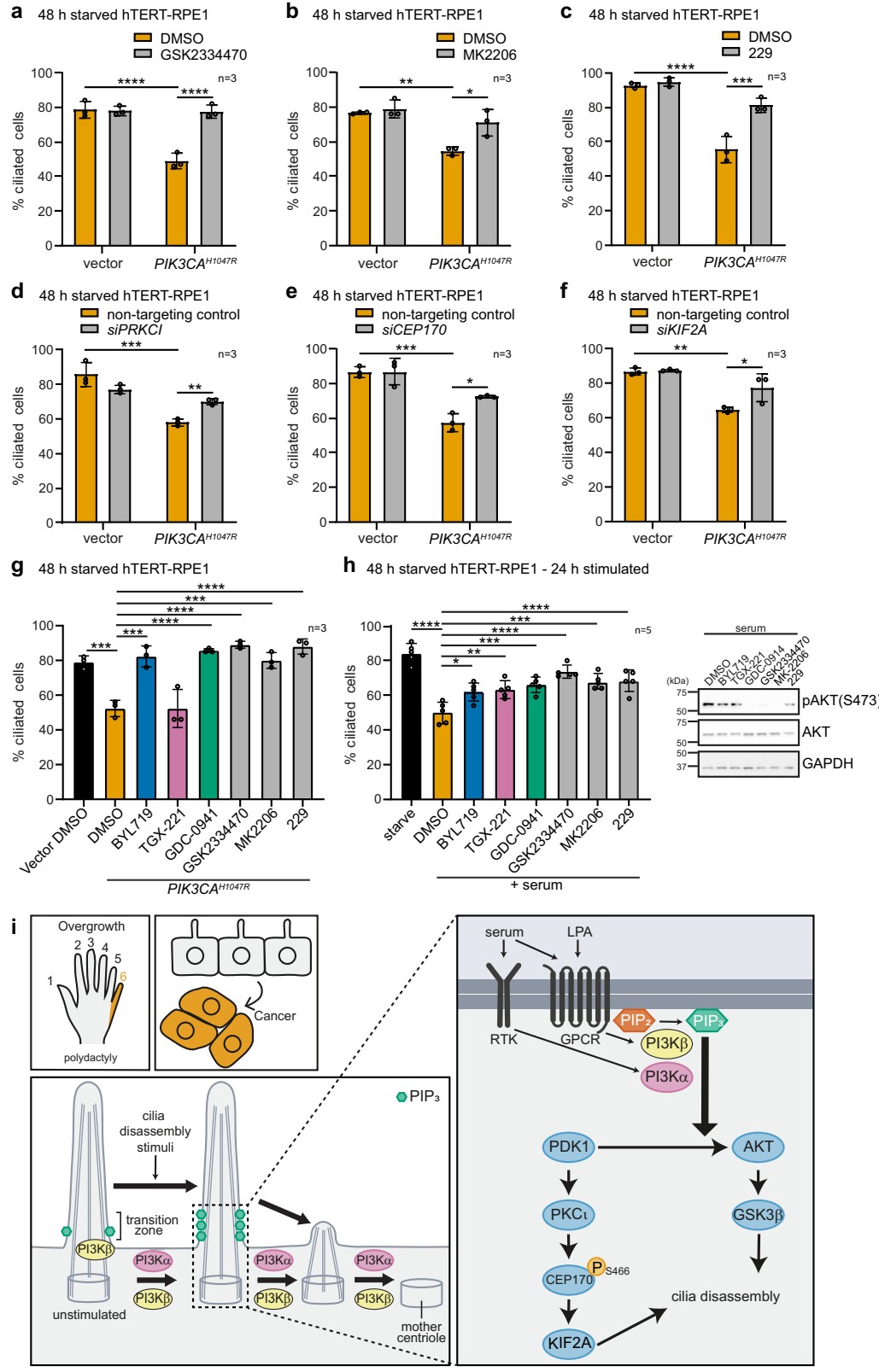

It is important to note that *Pik3ca^{H1047R}* embryos exhibit an earlier more severe phenotype compared to *Inpp5e*-null embryos and that PROS does not phenocopy Joubert or MORM syndrome with *INPP5E* mutations, even though both result in increased ciliary PIP₃. We hypothesize this is because PI3Kα hyperactivation alone is sufficient to surpass the PIP₃ threshold required to reduce cilia stability, whereas *Inpp5e* loss only induces cilia instability in combination with another stimuli,

with the *Inpp5e*-null phenotype instead being caused by mislocalisation of ciliary signalling receptors[23,26–28,35,121]. Furthermore, it is difficult to directly compare PROS with Joubert or MORM syndrome as PROS results from mosaic constitutive activation of PI3Kα[122], while Joubert and MORM are caused by ubiquitous partial loss-of-INPP5E-function[35,123].

Here, we propose *PIK3CA* mutant conditions may be classed as DCC as defined by Lovera et al.[3], namely as disorders in which "only a

**Fig. 7 | PDK1, PKCι, CEP170 and KIF2A drive cilia loss induced by PI3Kα activation.** hTERT-RPE1(Vector/*PIK3CA^H1047R*) cells were serum-starved for 48 h with doxycycline and treated ± (**a**) GSK2334470, (**b**) MK2206 or (**c**) 229 for the final 24 h, or transfected with non-targeted control, (**d**) *siPRKCI*, (**e**) *siCEP170* or (**f**) *siKIF2A*, fixed, stained with ARL13B and pericentrin antibodies and DAPI and the % ciliated cells scored, bars: mean ± SD, 100 cells scored/condition for *n* = 3 independent experiments, **p* < 0.05, ***p* < 0.01, ****p* < 0.001 *****p* < 0.0001 (two-way ANOVA (**a**) interaction *p* = 0.0002, row factor *p* = 0.0002, column factor p = 0.0003, (**b**) interaction *p* = 0.0336, row factor *p* = 0.0006, column factor *p* = 0.0092, (**c**) interaction *p* = 0.002, row factor *p* = $1.196 \times 10^{-5}$, column factor *p* = $7.744 \times 10^{-4}$, (**d**) interaction *p* = 0.0018, row factor $4.962 \times 10^{-5}$, column factor *p* = 0.4681, (**e**) interaction *p* = 0.0295, row factor *p* = $5.185 \times 10^{-5}$, column factor *p* = 0.0295, (**f**) interaction *p* = 0.0403, row factor *p* = 0.0002, column factor *p* = 0.0264). **g** hTERT-RPE1(Vector/*PIK3CA^H1047R*) cells were serum-starved for 48 h with

doxycycline and treated with BYL719, TGX-221, GDC-0941, GSK2334470, MK2206, 229 or DMSO. Cells were fixed, stained with ARL13B and pericentrin antibodies and DAPI and the % ciliated cells scored, bars: mean ± SD, 100 cells scored/condition for *n* = 3 independent experiments, ****p* < 0.001, *****p* < 0.0001 (one-way ANOVA, *p* = $3.217 \times 10^{-7}$). **h** hTERT-RPE1 cells were serum-starved for 48 h, pre-treated with BYL719, TGX-221, GDC-0941, GSK2334470, MK2206, 229 or DMSO for 1 h and then stimulated with serum in the presence of inhibitors for 24 h. Left, cells were fixed, stained with ARL13B and pericentrin antibodies and DAPI and the % ciliated cells scored, bars: mean ± SD, 100 cells scored/condition for *n* = 5 independent experiments, **p* < 0.05, ***p* < 0.01, ****p* < 0.001, *****p* < 0.0001 (one-way ANOVA, *p* = $2.092 \times 10^{-9}$). Right, cells were lysed and immunoblotted with pAKT(S473), AKT and GAPDH antibodies, blots representative of *n* = 3 independent experiments. **i** Model for PI3K/PIP₃-driven regulation of cilia disassembly. Source data are provided as a Source Data file.

subset of the clinical manifestations result from ciliary impairments which may be obscured by additional, non-ciliary phenotypes".

It has been proposed that this term is specifically relevant for mutant proteins localising outside the cilium/centrosome and thus having indirect effects on cilia function[3]. However, in light of new evidence, both of these additional classifications are troublesome, and we would recommend refraining from including these criteria in the definition of DCC for the following reasons. Firstly, regarding the point of subcellular localisation, many so-called non-ciliary proteins have now been identified at cilia, while 'classical' cilia-specific proteins have been found to have extra-cilia functions[3,123–125]. Secondly, assigning the function of a protein on cilia biology as 'direct' or 'indirect' is also largely open to interpretation. Specifically with regards to PI3K, this enzyme acts by phosphorylating a membrane lipid in cilia which then modulates the function of downstream effector proteins that impact cilia biology. PI3K could therefore be defined as having an 'indirect' effect on cilia. However, one could also argue PI3K has a 'direct' effect on cilia, given that the PI3K lipid substrates and effector proteins localise to the cilium. Therefore, in relation to DDC, but also to the cilia field in general, it would be beneficial to use more specific terms rather than 'direct/indirect' to describe the function of a protein at cilia.

We therefore propose to apply the definition of DCC, independent of the mutant protein sub-cellular localisation. This is clearly illustrated by the case of *Pik3ca^H1047R*-expressing embryos and PROS-affected individuals where polydactyly and cystic kidneys may result from cilia dysfunction, whereas limb overgrowth and vascular malformation result from non-ciliary functions.

Overall, our study adds an additional organelle to the class I PI3K repertoire which we speculate may account for some of the previously unexplained phenotypic consequences of PI3Kα over-activation.

## Methods

Mice were maintained at University College London according to UK The Animals (Scientific Procedures) Act 1986 Amendment Regulations 2012 (approved by the Animal Welfare and Ethical Review Body (AWERB), P434BB714 and PP5281579) and the CMCiB Animal Facility following the Catalan Ministry of Agriculture, Livestock, Fisheries and Food guidelines (protocols approved by CEEA Ethics Committees; animal Use Protocol number 9725).

### Antibodies and reagents

Antibodies: pAKT(S473) (IB 1:1000; #9271), pAKT(T308) (IB 1:1000; #9275), AKT (IB 1:1000; #9272), PI3Kα (IB 1:1000; #4249), pPRAS40(S246) (IB 1:1000; #2640), Ki67 (Alexa Fluor 488 Conjugate, IF 1:200; #11882), pS6RP(Ser240/244) (IF 1:800; #5364) from Cell Signalling Technologies (Danvers, MA, USA). ARL13B (IF 1:200; ab136648), AlexaFluor647-conjugated anti-ERG (retinal immunostaining 1:200; ab196149), GAPDH (IB 1:10,000; ab8245) and pericentrin, (IF 1:1600; ab4448) from Abcam (Cambridge, MA, USA). ARL13B (retinal immunostaining 1:100; 17711-1-AP) was from Proteintech (Rosemont,

IL, USA). CEP170 (IB 1:1000; IF 1:400; HPA042151), γ-tubulin (IF 1:200; T5326) and acetylated α-tubulin (IF 1:200-1:1000; T7451) were from Sigma-Aldrich (St. Louis, MO, USA). HA (IF 1:500; HA.11) from BioLegend (San Diego, CA, USA). PI(3,4)P₂ (IF 1:200; Z-P034) and PI(3,4,5)P₃ (IF 1:100; Z-P345b) from Echelon Biosciences (Salt Lake City, UT, USA). Mouse SMO (IF 1:100; sc-166685) was from Santa Cruz Biotechnology (Dallas, TX, USA). Alexa-Fluor-488/568/594/647-conjugated mouse and rabbit secondary antibodies (IF 1:600, retinal immunostaining 1:200) and AlexaFluor568-conjugated isolectin GS-B4 (retinal immunostaining 1:200, I21412) were from ThermoFisher Scientific (Waltham, MA, United States). Click-iT EdU cell proliferation kit for imaging (Alexa Fluor 594, C10339) was from ThermoFisher Scientific. HRP-conjugated mouse (NXA931V) and rabbit (NA934V) secondary antibodies (IB 1:5,000) were from Cytiva (Marlborough, MA, USA).

### Plasmid vectors

pSLIK-Neo was a gift from Iain Fraser (Addgene plasmid # 25735; http://n2t.net/addgene:25735; RRID:Addgene_25735)[126] (Watertown, MA, USA). pcDNA3.1-HA was a gift from Oskar Laur (Addgene plasmid # 128034; http://n2t.net/addgene:128034; RRID:Addgene_128034). pEN TMCS *PIK3CA^H1047R* and pSLIK-hygro *PIK3CA^H1047R* were from Robert Semple (University of Edinburgh). *PIK3CA^H1047R* was cloned into pSLIK-Neo using Gateway cloning according to the manufacturer protocol (ThermoFisher Scientific, 11791020). HA-CEP170 and HA-CEP170(S466D) were cloned into the pcDNA3.1-HA multiple cloning site using *Acc65*I and *Not*I restriction sites by GENEWIZ (Burlington, MA, USA). All plasmids were sequenced for verification.

### Cell culture

hTERT-RPE1 cells were purchased from ATCC (Manassas, VA, USA; CRL-4000) and cultured in Dulbecco's Modified Eagle Medium/F-12 with 10% FBS and 0.01 mg/ml hygromycin B (10687010, ThermoFisher Scientific) at 37 °C and 5% CO₂. hTERT-RPE1 cells were starved for the indicated time points in Dulbecco's Modified Eagle Medium/F-12. The authenticity of hTERT-RPE1 cells is tested by ATCC using short tandem repeat analysis, immunocytochemistry for pan-cytokeratin and flow cytometry for Ep-16 expression. Bulk frozen stocks were prepared immediately following receipt and used within 2 months of defrosting.

siRNA-mediated knockdown was performed using DharmaFECT transfection reagent (Tube 2) (T-2001-02, PerkinElmer, Waltham, MA, USA). hTERT-RPE1 cells were transfected with ON-TARGETplus siRNAs (Non-targeting Control Pool; D-001810-10-05, Human *PRKCI* siRNA; L-004656-00-0005, Human *CEP170* siRNA; 021258-00-0005 or Human *KIF2A*; 004959-00-0005, PerkinElmer) using DharmaFECT according to manufacturer specifications.

HEK293 (LentiX) cells were purchased from Clontech (Mountain View, CA, USA, NC9834960) and maintained in DMEM, supplemented with 10% FBS and 1% penicillin-streptomycin.

Transient transfection of HEK293 cells with pcDNA3.1-HA, HA-CEP170, HA-CEP170(S466D) plasmids was performed using

Lipofectamine 2000 (ThermoFisher Scientific) according to the manufacturer instructions.

*PIK3CA*[H1047R] lentiviruses were produced by co-transfecting LentiX cells with pSLIK-hygro *PIK3CA*[H1047R] or pSLIK-neo *PIK3CA*[H1047R] or empty vector plasmids and packaging vectors (pCMV-dR8.91, pCMV-VSV-G) using TransIT-LT1 according to the manufacturer instructions. 16 h post-transfection cells were treated with sodium butyrate (final concentration of 12.5 mM) for 6 h. The cells were then washed with PBS and media replaced. Following 20 h incubation, viral supernatant was collected and filtered through a 0.45 µm filter.

For generation of inducible *PIK3CA*[H1047R] hTERT-RPE1 cells, cells were seeded at 40–50% confluency the day prior to transduction. Cells were washed, the media replaced with DMEM/F-12 containing tetracyclin-free FBS (631101, Takara) and treated with lentiviral particles encoding pSLIK-neo *PIK3CA*[H1047R] or pSLIK-neo vector alone, in the presence of 20 µg/ml polybrene. Following 2 days incubation, cells were expanded, and transduced cells selected with 0.5 mg/ml G418 (10131-035, ThermoFisher Scientific) for 16 days. For experiments, empty-vector and *PIK3CA*[H1047R]-transduced hTERT-RPE1 cells were treated with 0.5 µg/ml doxycycline (Sigma-Aldrich, D9891-10G).

MCF10A cells were purchased from ATCC (CRL-10317) and cultured in DMEM/F12 containing 5% horse serum, 20 ng/ml EGF, 0.5 mg/ml hydrocortisone, 100 ng/ml cholera toxin, 10 µg/ml insulin, and 1% penicillin-streptomycin. For cilia formation, MCF10A cells were starved in DMEM/F12 containing 0.5 mg/ml hydrocortisone, 100 ng/ml cholera toxin and 1% penicillin-streptomycin.

For generation of inducible *PIK3CA*[H1047R] MCF10A cells, the cells were seeded at 10% confluency the day prior to transduction. Cells were washed, the media replaced with serum-free media and treated with lentiviral particles encoding pSLIK-hygro *PIK3CA*[H1047R] or pSLIK-hygro vector alone and 8 µg/ml polybrene. 6 h after transduction serum-containing media was added to cells which were then incubated for 2 days. Cells were then expanded and selected with 200 µg/ml hygromycin B (ThermoFisher Scientific, 10687010). 0.5 µg/ml doxycycline (Sigma-Aldrich, D9891-10G) was used to induce *PIK3CA*[H1047R] expression for experiments.

Immortalised *Pik3ca*[+/+] and *Pik3ca*[-/-] MEFs were generated and described previously[74]. MEFs were cultured in DMEM containing 10% FBS and 1% penicillin-streptomycin and starved in serum-free DMEM with 1% penicillin-streptomycin at 37 °C and 5% $CO_2$.

A549 cells were purchased from ATCC (CCL-185) and cultured in RPMI-1640 medium supplemented with NaPyr, 1% penicillin-streptomycin and 10% FBS. CRISPR/Cas9 was used to generate *PIK3CA* knockout A549 cells[75]. Confirmed *PIK3CA* knockout and wild-type clones were used to produce *PIK3CA* knockout and wild-type pools which were used for experiments.

BPH1 cells were purchased from DSMZ (ACC 143, ref.14602) and cultured in DMEM supplemented with 10% FBS and 1% penicillin-streptomycin at 37 °C and 5% $CO_2$. Cells were treated with 5 µM BKM120 for 8 h as indicated in the figure legend.

Cilia assembly/disassembly assays were performed using standard protocols[18,33,34]. Cells were serum-starved for 48 h, followed by stimulation with serum or 2 µM LPA for up to 24 h as indicated in the figure legends. To test the impact of inhibitors on cilia disassembly, cells were pre-treated with 0.25 µM BYL719 (S2814; Selleck Chemicals, Houston, TX, USA), 0.5 µM TGX-221 (HY-10114; MedChemExpress, Monmouth Junction, NJ, USA), 0.5 µM GDC-0941 (S1065-SEL-10 mM/1 ml; Selleck Chemicals), 5 µM GSK2334470 (S7087; Selleck Chemicals), 2 µM MK-2206 (S1078; Selleck Chemicals), 2 µM 229 (obtained from Cancer Research Horizons and is structurally closely related to CRT0329868[127], manuscript in preparation), 0.2 µM parsaclisib (HY-109068; MedChem Express), 0.1 µM IPI-549 (S8330; Selleck Chemicals) or 1 µM GSK2636771 (A11784, Adooq Bioscience, Irvine of CA, USA) for 1 h, followed by stimulation with serum or 2 µM LPA (L7260, Sigma-Aldrich) for 24 h in the presence of inhibitors. For

PI3Kα activator-induced cilia disassembly, 48 h serum-starved cells were treated with 5 µM 1938[75] for up to 24 h as indicated in the figure legends. MCF10A cells were starved for up to 120 h to induce cilia assembly.

All cell lines were regularly tested for mycoplasma contamination.

## Mouse strains

All mice were on the C57BL/6 background. Mice were maintained in specific pathogen-free conditions and individually-ventilated cages with a 07:30–19:30 light and 19:30-07:30 dark cycle at 18–22 °C and 40–60% humidity. The age of embryos and pups used for experiments is indicated in the figure legends. Male and female mice were used for experiments. Sex based analysis was not performed as embryos were analysed up to E10.5 which is the bipotential stage.

*Pik3ca*[tm1.1Waph] (hereafter *Pik3ca*[H1047R]) mice were generated and described previously[41] and provided by Wayne Philips (Peter MacCallum Cancer Centre, Melbourne, Australia). *Tg(CMV-cre)1Cgn* (hereafter *CreDel*) mice ubiquitously expressing Cre recombinase from the zygote stage were generated and described previously[128]. Experimental embryos were derived from timed matings between *Pik3ca*[H1047R] and *CreDel* mice. Experimental embryos were heterozygous for both *Pik3ca*[H1047R] and *CreDel* transgenes. Embryos positive for *CreDel* were used as controls.

*Pik3ca*[H1047R] mice were also crossed with *Tg(Pdgfb-icre/ERT2,-EGFP) 1Frut* (hereafter *Pdgfb-CreER*[T2]) mice expressing inducible *CreER*[T2] from the endogenous *Pdgfb* locus (endothelial cell-specific)[129]. Recombination was induced at P1 by intraperitoneal injection of 25 mg/kg 4-hydroxytamoxifen. *Pik3ca*[H1047R] mice treated with 4-hydroxytamoxifen were used as controls.

Yolk sacs were used for genotyping of embryos and ear notches were used for genotyping of postnatal mice via PCR analysis of genomic DNA using the following primers: For *Pik3ca*[H1047R], PIK3CA-Lat-20F – TTGGTTCCAGCCTGAATAAAGC, PIK3CA-Lat-20 – GTCCAAGGCTA GAGTCTTTCGG, PIK3CA-Lat-19F – TCCACACCATCAAGCAGCA for *CreDel*, Cre-1 – AGATGTTCGCGATTATCTTCTA, Cre-2 – AGCTACACC AGAGACGG, Actin-F – GGTGTCATGGTAGGTATGGGT, Actin-R – CGCA CAATCTCACGTTCAG, and for *Pdgfb-CreER*[T2] PdgfbCre-F – CCAGCCG CCGTCGCAACT, PdgfbCre-R – GCCGCCGGGATCACTCTCG.

## Indirect immunofluorescence

Cultured cells were fixed with 4% paraformaldehyde for 20 min. Cells were permeabilized with −20 °C methanol for 5 min or 0.1% Triton X-100 in PBS for 90 s (for SMO IF) and blocked in 1% BSA for 30 min. Primary antibodies diluted in 1% bovine serum albumin were incubated for 1 h at room temperature. Cells were then washed three times with PBS and incubated with Alexa Fluor-conjugated secondary antibodies diluted in 1% BSA for 45 min at room temperature. Cells were washed and mounted using ProLong Gold antifade reagent with DAPI (ThermoFisher Scientific, P36935).

For cilia staining of BPH1 cells, after washing 3 times with cold PBS, cells were fixed with 4% PFA supplemented with 0.1% Triton X-100 in PBS for 15 min at RT. Then, cells were washed 3 times with PBS. Blocking was performed for 1 h at 37 °C in blocking buffer (BB: 2% serum, 1% BSA in PBS). Primary antibodies were incubated overnight at 4 °C and cells were washed with PBS 3 times. Secondary antibodies were incubated for 1 h at 37 °C, followed by nuclear staining with DAPI (10 min, 500 ng/ml in PBS; Sigma).

$PI(3,4)P_2$ immunofluorescence was performed using the 'Golgi' PI staining protocol as described previously[130].

## EdU cell proliferation assay

Cells were cultured as appropriate and labelled with 10 µM EdU for 4 h. The cells were then fixed, stained with Click-iT EdU Alexa Fluor 594 Imaging Kit (Thermofisher Scientific, C10339) according to manufacturer specifications and mounted using ProLong Gold antifade

reagent with DAPI. Dynamic range of assay and labelling time was confirmed by comparison of EdU labelling of serum starve vs serum stimulated cells.

## Mouse retina isolation and immunostaining

Mice were sacrificed by decapitation and eyes were isolated, followed by 1 h incubation on ice in 4% PFA in PBS. Isolated retinas were fixed for an additional 1 h, permeabilised overnight at 4 °C in permeabilisation/blocking buffer (1% BSA, 0.3% Triton X-100 in PBS). Afterwards, the retinas were incubated overnight at 4 °C with ARL13B antibodies diluted in permeabilisation/blocking buffer. Samples were washed three times in PBS containing 1% Tween-20 (PBST), following incubation with PBlec buffer (1% Triton X-100, 1 mM CaCl$_2$, 1 mM MgCl$_2$ and 1 mM MnCl$_2$ in PBS, pH 6.8) for 30 min at room temperature. AlexaFluor647-conjugated ERG and AlexaFluor488-conjugated rabbit antibodies, diluted in PBlec, were added to the retinas and incubated for 2 h. Blood vessels were visualised with AlexaFluor568-conjugated isolectin GS-B4. Following three washes with PBST, the retinas were flat-mounted on a microscope slide and imaged by confocal microscopy.

## Microscopy

Microscopy was performed at University College London, Cancer Institute (London, UK) or Center for Cooperative Research in Biosciences (CIC BioGUNE), Basque Research and Technology Alliance (Derio, Spain) at room temperature. Confocal microscopy was performed using a Zeiss (Oberkochen, Germany) LSM 880 microscope with Airyscan with a ×63 oil Plan-Apochromator 1.4 numerical aperture objective lens, PMT detector, 405 nm, 458 nm, 488 nm, 514 nm, 561 nm, 594 nm and 633 nm lasers and Zen Black acquisition software with 512 x 512 pixels/image. Widefield microscopy was performed using a Zeiss Z1 Inverted microscope with a 100x oil objective lens, DAPI, GFP, Cy5, PI and TxRed fluorescent cubes and Zen Blue acquisition software or an upright wide-field fluorescent microscope (Axioimager D1, Zeiss, with Zen acquisition software). Stereo microscopy was performed using a Leica M205FA stereo microscope with Leica Application Suite acquisition software. Image processing was performed using Fiji ImageJ software (National Institutes of Health, USA) and was limited to alterations of brightness, subjected to the entire image. For presentation images were false coloured using linear lookup tables covering the full range of the data.

## Image analysis

Ciliary PI MFI was measured using Fiji ImageJ in hTERT-RPE1 cells and MFEs. Cells were immunostained with PI antibodies and ARL13B antibodies to define the ciliary axoneme and imaged by confocal microscopy at the same laser scanning intensity and Airyscan processing. A macro was developed to identify the cilia axoneme using the ARL13B channel, skeletonize the signal and define the location. Boxes of standardised size were then drawn at either end of the axoneme centred around the highest intensity PI pixels. The PI MFI was measured in each box and the MFI for the highest intensity box, representing the cilia base, returned. The PI MFI was plotted as a histogram using R.

pS6RP(S240/244) MFI was measured using Fiji ImageJ in hTERT-RPE1 cells. Cells were immunostained with pS6RP(S240/244) antibodies and ARL13B antibodies to define the ciliary axoneme and imaged by confocal microscopy as z-stacks at the same laser scanning intensity. A region of interest was defined by outlining the cell and excluding the nucleus and the MFI measured in the region of interests. The pS6RP(S240/244) MFI was plotted as a histogram using R.

Ciliary SMO MFI was measured using Fiji ImageJ in hTERT-RPE1 cells. Cells were immunostained with SMO and acetylated tubulin antibodies and imaged by confocal microscopy at the same laser scanning intensity and Airyscan processing. A region of interest was defined by outlining the acetylated tubulin demarked axoneme and the SMO MFI measured in the region of interests. The SMO MFI was plotted as a histogram using R. To measure cilia length, cells were stained using ARL13B antibodies to define the ciliary axoneme and imaged by confocal microscopy as z-stacks at the same laser scanning intensity. Images were analysed using Imaris (Oxford Instruments, Abingdon, UK) with the filament tracer tool to measure the length of each cilium in 3D.

## Cell cycle analysis

For cell cycle analysis of MEFs, cells were synchronised in medium (DMEM) with low (0.5%) FBS content for 16 h and then transferred in medium containing 10% serum. Cell cycle distribution of MEFs was assessed after 48 h by FACS analysis of BrdU incorporation using a FITC BrdU Flow Kit (559619; BD Pharmingen) according to the manufacturer instructions. Data analysis was performed using Summit V4.0 (DakoCytomation). For cell cycle analysis of BPH1, cells were collected in PBS, fixed in 4% PFA for 15 min at room temperature and permeabilized with 0.2% Triton X-100 in PBS for 30 min at room temperature. Then, pelleted cells were stained with propidium iodide solution (2% PI and 0.1 mg/ml RNAse A in PBS) for 30 min at 37 °C and analysed by FACS (BD Biosciences). Data analysis was performed using FlowJo v10.10.

## Mass spectrometry-based discovery phosphoproteomics

hTERT-RPE1 cells, cultured in 15 cm dishes, were serum-starved for 48 h and stimulated for experiment 1 by the addition of serum or 2 µM LPA in the presence or absence of 0.5 µM GDC-0941 or 0.05% DMSO control for 15 min or 2 h. For experiment 2, cells were starved for 24 h and stimulated with 5 µM 1938 or 100 nM insulin (I5016; Sigma-Aldrich) in the presence or absence of 0.25 µM BYL719 or 0.025% DMSO control for 15 min or 4 h. 500 µl of urea lysis buffer [50 mM triethylammonium bicarbonate, 8 M urea, cOmplete™, EDTA-free protease inhibitor cocktail (1:50 dilution) (11873580001; Roche), 1 PhosSTOP tablet (4906845001; Roche) per 10 ml of lysis buffer, 1 mM sodium orthovanadate] was used to lyse cells and the lysate sonicated for 11 min with breaks on ice (alternating 45 s on, 45 s off). Protein concentration was determined using the BCA protein assay (23227; Pierce) and samples were normalised to contain 300 µg of protein in an equal volume of buffer. Proteins were reduced with 5 mM Tris(2-carboxyethyl)phosphine hydrochloride (C4706; Sigma-Aldrich) at 37 °C for 20 min, followed by 10 mM 2-chloroacetamide (22790; Sigma-Aldrich) alkylation for 20 min at room temperature (protected from light). LysC (129-02541; FUJIFILM Wako Chemicals, Osaka, Japan) protein digestion was performed at a protease to protein ratio of 0.00867 (experiment 1) or 0.012 (experiment 2) for 3.5 h at 37 °C. The urea concentration of the buffer was diluted to 1.5 M via the addition of 50 mM triethylammonium bicarbonate (T7408; Sigma-Aldrich) to samples. Peptides were then digested with trypsin (V5113; Promega) at a protease to protein ratio of 0.02 overnight at 37 °C. Digests were quenched using 10% trifluoroacetic acid (302031-M; EMD Millipore) to a final pH of 2.0. Samples were desalted using 35–350 µg C18 columns (HMM S18V; The Nest Group, Inc., Southborough, MA, USA) according to the manufacturer specifications. For experiment 1, zirconium IMAC (MagReSynZr-IMAC HP MR-ZHP002, ReSyn Biosciences, Gauteng, South Africa) was used for phosphoenrichment according to manufacturer protocol, using a bead to peptide ratio of 4:1. For experiment 2, TiO$_2$ (5020-75010; Hichrome Titansphere TiO$_2$, 10 µm capacity, 100 mg, GL Sciences) phosphoenrichment was performed, briefly peptides were loaded onto TiO$_2$ beads and washed with 1 M glycolic acid (124737; Sigma-Aldrich)/80% acetonitrile/5% trifluoroacetic acid followed by 80% acetonitrile/0.2% trifluoroacetic acid, 20% acetonitrile and elution with 5% ammonium hydroxide. Following phosphoenrichment sample desalting was performed using 7–70 µg C18

columns (HUM S18V; The Nest Group, Inc., Southborough, MA, USA) according to the manufacturer specifications. Phosphopeptides were dried, stored at −80 °C and resuspended in 10% formic acid prior to analysis. nLC-MS/MS was performed on a Orbitrap Exploris 480 coupled to an Easy-nLC 1200 (Thermo Scientific). Fifty percent of each sample was analysed as 10 µl injections. Peptides were loaded on a 25 cm (75 µm ID, 1.7 µm 120 Å pore size C18 resin) Aurora Ultimate UHPLC packed emitter column (Ion Opticks) housed in a Nanospray Flex Ion Source modified to include a column oven (Sonation GmbH) set to 35 °C. Peptides were separated using a linear gradient from 6% to 38% of buffer B (buffer A: 0.1% formic acid in water, buffer B: 80% acetonitrile/0.1% formic acid) over 120 min, at a flow rate of 250 nl/min. Peptides were ionised by electrospray ionisation using 1.8 kV applied immediately prior to the analytical column via a microtee built into the nanospray source with the ion transfer tube heated to 320 °C and the RF-lens set to 45%. Precursor ions were measured in a data-dependent mode in the orbitrap analyser scanning between 375–1200 m/z at a resolution of 120,000 and a normalised AGC target of 300% (3e6 ions) (max. MS1 injection time set to "Auto"). With the duty cycle fixed to 3 s, precursors were subjected to MS/MS fragmentation under the following criteria – Monoisotopic Peak Determination set to "Peptide", Intensity threshold set to 1e5, Charge states set to "2-5". MS/MS scans were generated using HCD fragmentation with collision energy set to 30% and measured in the orbitrap at a resolution of 15,000 and a normalised AGC target of 100% (1e5 ions) (max. MS2 injection time set to "Auto").

### Peptide identification, quantification and statistical analysis of discovery phosphoproteomics data

Raw data were analysed with MaxQuant[131] (version 1.6.17) where they were searched against the human SwissProt database [http://www.uniprot.org/] (downloaded 08/03/2022) using default settings. Carbamidomethylation of cysteines was set as fixed modification, and oxidation of methionines, acetylation at protein N-termini, phosphorylation (on S, T or Y) were set as variable modifications. FTMS MS2 mass tolerance is set to 20 ppm, the software carries out an automatic mass calibration. Enzyme specificity was set to trypsin with maximally 2 missed cleavages allowed. Minimum peptide length was set to 7 amino acids. To ensure high confidence identification, peptide-spectral matches, peptides, and proteins were filtered at a less than 1% false discovery rate (FDR). Label-free quantification in MaxQuant was used with a LFQ minimum ratio count of 2, Fast LFQ selected and the 'skip normalisation' option selected. The 'match between runs' feature was selected with a match time window of 0.7 min and an alignment time window of 20 min. The MaxQuant 'phospho(STY) Sites.txt' output file was reformatted by merging each protein accession and gene name with its corresponding phosphosite to obtain an 'Annotated_PhosphoSite.txt'. This file, together with the MaxQuant 'evidence.txt' output file and an experimental design 'annotation.csv' file, was further processed by removing contaminants and reversed sequences, and the removal of phosphosites with 0 or 1 valid values across all runs. High experimental reproducibility was observed, as evidenced by an average Pearson Correlation Coefficient for experiment 1 of $r = 0.87$ (Supplementary Fig. 2a) and experiment 2 of $r = 0.82$ for biological replicates (Supplementary Fig. 6c). Quantified phosphopeptides were analysed within the model-based statistical framework MSstats[132] (version 3.20.0, run through RStudio (version 1.3.1093, R version 4.0.3)). Data were log2 transformed, quantile normalised, and a linear mixed-effects model was fitted to the data. The group comparison function was employed to test for differential abundance between conditions. $p$-values were adjusted to control the FDR using the Benjamini-Hochberg procedure[133]. The mass spectrometry proteomics data have been deposited to the ProteomeXchange Consortium via the PRIDE[134] partner repository with the dataset identifier PXD046462.

### Kinase substrate enrichment analysis

KSEA was performed as described previously[72] using the OmniPath[71], Edges[72] and PhosphoSitePlus[73] databases. Scripts are at github/CutillasLab.

### Immunoblot analysis

Cells were washed twice in ice-cold PBS and lysates were harvested in lysis buffer (0.05 M Tris pH 7.4, 0.1 M NaCl, 0.05 M NaF, 0.005 M EDTA, 1.97 mM EGTA, 1% Triton X100 with phosphatase (524625-1SET; Merck) and protease (539131-10Vl; Merck) inhibitors). Following 20 min extraction on ice, homogenates were centrifuged at 20,800 g at 4 °C for 10 min and supernatant extracted. Protein concentration was determined using the Bradford assay (500-0006, BioRad, Hercules, CA, USA). Lysates were resolved by SDS-PAGE, transferred to polyvinylidene fluoride membranes (IPVH00010; Sigma-Aldrich) using wet transfer (BioRad) or nitrocellulose membranes (IB23001X3; Thermo-Fisher Scientific) using dry transfer (iBlot2, ThermoFisher Scientific). Membranes were blocked and 5% BSA, incubated with primary antibodies overnight at 4 °C and secondary antibodies at room temperature for 1 h before detection by chemiluminescence (WBLUF0500, Immobilon Forte Western HRP substrate Merck) using a Gel Doc (Bio-Rad).

### Quantitative reverse transcription PCR

RNA was isolated from embryos and cell lines with the Qiagen RNeasy mini-kit (Qiagen, Hilden, Germany). cDNA synthesis was performed from 50 ng of RNA using the iScript cDNA synthesis Kit (BioRad) according to the manufacturer instructions. cDNA samples were diluted 1:10 and analysed by qPCR using GoTaq® qPCR Master Mix (Promega, Madison, WI, USA) and QuantiTect (249900, Qiagen) *Ptch1, GLI1, Gli1, Axin2, Ccnd1, PRKCI, GAPDH, Gapdh, Actb* and *KIF2A* primers with a QuantStudio 5 Real-Time PCR System (ThermoFisher Scientific). All samples were analysed in triplicate and expression normalized to *GAPDH* or *Gapdh* as appropriate using the ΔΔCt method[135]. Key results were also validated using *Actb* as a house keeping gene (not shown).

### ADP-GloTM kinase assay

Peptide kinase assays were performed using an ADP-Glo kit (Promega) as per manufacturer instructions. In each 5 µl reaction 20 ng of PKCι kinase domain purified to homogeneity (2 P, phosphorylated at activation loop and turn motif priming sites) was incubated with excess amount of peptide (β-PSS: CONH2-ESTVRFARKGSLRQKNVH-CONH2, CEP170: CONH2-SIPFLRTALLRSSGSRGHR-CONH2, 250 µM) in a buffer containing 50 mM Tris, pH 7.4, 10 mM MgCl$_2$. Reactions were started with ATP (100 µM) and incubated at 32 °C for 60 min. The reaction was stopped by addition of 10 µl ADP Glo reagent and incubated for 40 min at room temperature. Next, 10 µl kinase detection reagent was added and the samples incubated for 30 min at room temperature. Luminescence was read out from an OptiPlate 384 well plate in an EnSight multimode plate reader (Perkin Elmer).

### Statistics & reproducibility

Statistical analysis was performed using GraphPad Prism 8.0 or 10.1.2. Graphs represent mean ± SD. Differences between groups were considered statistically significant when $p < 0.05$. All statistical tests used are indicated in the figure legends. In all cases two-sided tests were used. For Student's $t$ tests, sample variance was assessed by the F test to determine whether to use Welch's correction for unequal variance. For multiple comparisons one-way or two-way ANOVAs followed by Tukey's post hoc test were used as indicated in the figure legends (difference in sample variance assessed by the Brown–Forsythe test). For comparisons of histogram distributions, Kolmogorov-Smirnov test or Kruskal-Wallis test was used as appropriate and indicated in the figure legends. The sample size is indicated in the figures and figure legends. Sample sizes were chosen based on our experience and

standards in the field, no statistical method was used to predetermine sample size. Statistical analysis used for phosphoproteomics data is described above and indicated in the figure legends. No data were excluded from the analyses. Mouse embryos were randomised based on their genotype and investigators were blind to embryo genotype during analysis.

## Reporting summary

Further information on research design is available in the Nature Portfolio Reporting Summary linked to this article.

## Data availability

Mass spectrometry data (raw data and processed data) generated in this study have been deposited to the ProteomeXchange Consortium via the PRIDE partner repository[134], with the dataset identifier PXD046462. The human SwissProt database [http://www.uniprot.org/], was used for analysis. The cell biology and in vivo data generated in this study are provided in the Source Data file. The other data that support the findings in this study are available from the corresponding author upon request. Source data are provided with this paper.

## Code availability

KSEA scripts are at the Cutillas Lab Github [http://www.github.com/CutillasLab].

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

## Acknowledgements

Personal funding was from the European Commission (H2020-MSCA-IF-2018 GA: 838559; to S.E.C.). Research in the B.V. laboratory was supported by BBSRC (BB/W007460/1), Cancer Research UK (C23338/A25722) and PTEN Research (UCL-20-001) (B.V.). La Caixa Banking Foundation Junior Leader project (LCF/BQ/PR20/11770002) (S.D.C.). The Spanish Association Against cancer (AECC) (L.B.). The AECC (GCTRA18006CARR) and the European Research Council (Consolidator Grant 819242) (A.C.). CIBERONC was co-funded with FEDER funds and funded by ISCIII (A.C.). The UCL Cancer Institute Translational Technology Platforms are supported by the CRUK City of London Centre Award (CTRQQR-2021\100004) (S.S. and A.B.). We acknowledge funding from the Francis Crick Institute, which receives its core funding from Cancer Research UK (CC2068), the UK Medical Research Council (CC2068) and the Wellcome Trust (CC2068) (P.J.P., N.Q.M. and M.C.). R + D + € project PID2020-116184RB-I00 funded by MCIN/AEI/10.13039/501100011033 for work performed in the M.G. laboratory. We thank the UCL Cancer Institute Microscopy and Imaging Translational Technology Platform, Proteomics Research Translational Technology Platform and Genomics Translational Technology Platform. We also thank the UCL Research Capital Infrastructure Fund (RCIF) and the National Institute for Health Research University College London Hospitals Biomedical Research Centre for upgrade of the UCL Cancer Institute Microscopy facility, Priyanka Tibarewal for excellent feedback on the manuscript and Tchern Lenn for help developing the macro for cilia phosphoinositide quantification.

## Author contributions

S.E.C. and B.V. provided the initial study conceptualization, supervised the study and wrote the manuscript with contributions from all authors. S.E.C, W.P., A.B., L.C.F., M.C., S.S., L.B. and S.D.C. designed and performed experiments. M.A.D. and M.A. performed experiments. S.E.C., A.B., S.S. and P.R.C. analysed data. B.B. generated mutant cell line models. M.G., N.Q.M., P.J.P. and A.C. provided scientific input and key reagents.

## Competing interests

B.V. is a consultant for iOnctura (Geneva, Switzerland), Pharming (Leiden, the Netherlands) and shareholder of Open Orphan and Poolbeg Pharma (Dublin, Ireland), P.J.P. is a consultant for Apollo Therapeutics and Phoremost. P.R.C. is a co-founder and director of Kinomica Ltd, a company active in the area of phosphoproteomic diagnostics. The remaining authors declare no competing interests.
