## [Peer Review File · Nature Communications]

A class I PI3K signalling network regulates primary cilia disassembly in normal physiology and diseaseREVIEWER COMMENTS

Reviewer #1 (Remarks to the Author):

In this manuscript, a variety of pharmacologic and genetic tools are used to probe the role of Class I PI3K activation in the process of cilia disassembly.

Firstly, a mouse model of constitutive PI3K α activity is shown to have defects in ciliary signaling and a reduction in ciliated cells in the retina pigment epithelium (RPE). Using immortalized RPE cells, serum or LPA stimulation is shown to activate the PI3K pathway, and its inhibition reduces cilia disappearance after adding these stimuli. PI3K lipid products are shown to be present in the transition zone of primary cilia and to be synthesized by PI3K β in RPE cells, but in murine embryonic fibroblasts, EGF stimulation increases PIP3 through both PI3K α and β . Constitutively active PI3K α is shown to induce cilia disassembly in RPE cells, as does pharmacologic activation of the enzyme. Loss of PI3K activity, conversely, increases the proportion of ciliated cells in two lines (models for prostate hyperplasia and lung cancer). PI3K α activation leads to phosphorylation of S466 in centriolar protein CEP170, and phosphomimetic mutants of this protein similarly promote cilia disassembly in HEK293 cells. Finally, using knockdown or pharmacologic inhibition, requirements for PDK1, PKC ζ and CEP170 are shown to be required for PI3K α -induced cilia disassembly.

Overall, the manuscript is clearly written, and the experiments produced robust data that are clearly presented. There are two points of novelty here: firstly, a mechanism whereby PI3K activation can lead to cilia-loss associated phenotypes, both in PROS and possibly in cancer or neoplasms. Secondly, a potentially novel signaling axis via PKC ζ to CEP170. These are conceptual advances in the fields of cilia biology and disease as well as in signal transduction and PI3K signaling. Therefore, the manuscript justifies publication in a wide focus and broadly disseminated journal like Nat. Comms.

Overall, I was convinced by the data and believe the story is largely complete. That said, I did note some specific areas where the manuscript could be strengthened:

(1) Requirements of PI3K for cilia disassembly: treatment of cells with serum seems to be the most robust stimulus for depleting cilia and yet, from the data presented in figures 3 and 7, blockade of the pathway does not completely block cilia disassembly in all cells. Notably, the statistical comparison of inhibitor treatment to starved cell control is not presented, but there does still appear to be a difference even in GDC-0941 treated cells. This is addressed in the manuscript with the statement that "PI3K and downstream signalling is only one, but a physiologically critical, component of the serum-induced cilia disassembly network". The reviewer agrees that these incomplete but nonetheless significant effects could disrupt cilia-associated physiology to a phenotype-inducing degree. However, these subtleties should be more clearly addressed in the discussion.

(2) Localization of the PI3K-induced signal leading to cilia disassembly: Given the identification of CEP as a downstream effector of the pathway, and the long established roles of phosphoinositides up- and down-stream of PI3K at primary cilia, the presence of

PIP3 at the transition zone reported in figure 4 makes sense. The dependence solely on PI3Kbeta for a resting pool is surprising but convincing. Here, EGF induced increases in ciliary PIP3 are convincingly shown to require both PI3K alpha and beta in embryonic fibroblasts, which seem to have been selected due to the ability to use Pik3ca null lines. However, the majority of the manuscript explores the effects of serum (and to a lesser extent, LPA) on cilia disassembly in RPE1 cells. Therefore, the manuscript would be substantially strengthened if effects of 1938, or serum \pm PI3K inhibitors on ciliary PIP3 were shown in RPE-1 cells. This would unambiguously assign this pool of PIP3 as the pool that activates the cilia disassembly pathway.

(3) Robustness of the PIP3 localization data shown in figure 4: comparing the control conditions between RPE cells in 4a and 4c, the cells in figure 4c seem to have less PIP3 than the cells in figure 4a, perhaps even as low as cells in figure 4a treated with TGX-211. This undermines the robustness of the results somewhat. The data might be strengthened by plotting them differently, perhaps as a “super-plot” (doi: 10.1083/jcb.202001064) indicating the value of each cell in a scatter plot with experimental means indicated and the individual experiments color coded. If the experimental replicates are overlapping in each case and the difference between 4a and 4c is due to some other difference in experimental protocol, e.g. transfection vs not, this should be apparent. This will also eliminate any skew from an outlier experiment.

Technical points:

(4) Extended figures 2 and 5 are posted at a resolution that is too low to really examine and interpret the data.

(5) Figure 3g left and right panels. This panel is a somewhat tricky game of spot the difference. The reader might benefit from a more accentuated highlighting of the 24 h serum vs 48 h serum in the two panels.

(6) Many figure panels rely on scoring the % ciliated cells. The data are correctly reported as $n = 1$ per experiment. However, the reviewer did not find an indication of how many cells were scored per experiment. This would be beneficial for assessing the robustness of the score.

Reviewer #2 (Remarks to the Author):

This manuscript described a transgenic mouse line carrying Pik3ca H1047R/+ that showed embryonic lethality at E9.5 with some phenotypes like some of ciliopathy phenotypes. The authors suggested that these developmental abnormalities were caused by PI3Ka/b-induced cilia instability. It is appreciated that unbiased proteomics analyses were performed to study the global changes following hyperactivation of PI3Ka. However, PI3Ka and PI3Kb are both ubiquitously expressed in cell and activating either of them triggers numerous downstream signaling events including the ones that facilitate cell proliferation and indirectly promote

cilia disassembly. The lack of clear mechanistic evidence demonstrating the direct, cilium-specific role of PI3Ka/b severely dampened the significance of their findings.

1. PI3K-H1047R somatic mutation in human causes significant overgrowth, leading to hemihypertrophy and malformation of the vascular system. Polydactyly was observed in CLOVE patients but rarely. No progressive ciliopathy phenotypes such as progressive polycystic kidneys is observed in patients. It is far-fetched to conclude that the pathogenic mechanism of PI3Ka-H1047R is that it causes primary cilia instability.

2. INPP5E is specifically localized in primary cilia. Inhibiting INPP5E also causes the increase of PIP3 in the transition zone area and decreases the stability of primary cilia. However, the INPP5E-null mice don't share similar phenotypes as the PIK3ca H1047R/+ mice. MORMS (mutated INPP5E not localizing to primary cilia) and JBTS1 (mutated INPP5E with less phosphatase activity) patients, also don't show similar phenotype as patients expressing PI3K-H1047R.

3. It is interesting that PIP3 level in the transition zone changed when PI3Ka/b was activated or inhibited. However, no studies were conducted to analyze the direct consequence of this PIP3 change in the context of primary cilia. For example, was INPP5E in cilia or the PI4P-PI4,5P2 distribution along the cilium changed? What about other structural, trafficking, and signaling molecules residing/functioning in cilia? Moreover, checking the accumulation of Gli3 at the ciliary tip and decrease of ciliary GPR161 induced by SAG is much more relevant in this scenario than checking the global transcription level of Gli1. Subsequently, if this increase of PIP3 is diminished/blocked specifically at the transition zone, would any ciliary changes (including the increased cilia instability) be inhibited?

4. Another very interesting and important question derived from the observation of PIP3 changes at TZ in response to PI3Ka/b activity: if PI3Ka/b is not specifically activated at the TZ, how could PIP3 at the TZ be changed? What is the intermediate player(s) transferring PI3K activity/product from the cytosol/plasma membrane to the TZ?

5. The observation that CEP170 can be phosphorylated at S466 provides an interesting lead for mechanistic analysis, which is lacking in the current manuscript. The HEK293 cell line is not a good model to study primary cilia.

6. Whether/how the AURKA signaling pathway is influenced by modifying PI3Ka/b activity and the subsequent influence on cilia stability should be determined.

7. Data presentation in Fig. 4 is confusing. What MFI does each of the red dots (PIP3 or PI(3,4)P2) in the images correlate to as shown in the corresponding histogram? Would the difference still significant if comparing the mean MFI of all cells quantified in each experimental condition?

Reviewer #3 (Remarks to the Author):

There are some comments on the technical side of phosphoproteomics.

1. Phosphosite Screening Process

The manuscript currently does not elaborate on the selection criteria for the phosphosites analyzed in the study. It is imperative to understand the basis on which these were chosen for subsequent analyses. Could you please provide a more detailed description of the screening process, including any statistical cutoffs, conservation analysis, or functional relevance that guided your selection?

2. Coefficient of Variation (CV) Absence

The manuscript does not currently report the coefficient of variation (CV) for the experimental conditions. The CV is a crucial statistical measure that can help ascertain the reproducibility and reliability of the observed results. I recommend that the authors include the CV calculations for each condition or provide a rationale for its omission.

3. Choice of DDA over DIA

The use of a data-dependent acquisition (DDA) approach, particularly with a long gradient, raises questions about the efficiency of the mass spectrometry run. Given that the 150-minute gradient was not fully utilized, could you discuss the reasons behind opting for DDA rather than data-independent acquisition (DIA), which may offer higher data integrity and depth? It would be beneficial for readers to understand the specific advantages that DDA provided for your study's objectives, as well as any considerations given to the use of DIA.

Reviewer #1:

In this manuscript, a variety of pharmacologic and genetic tools are used to probe the role of Class I PI3K activation in the process of cilia disassembly.

Firstly, a mouse model of constitutive PI3K α activity is shown to have defects in ciliary signaling and a reduction in ciliated cells in the retina pigment epithelium (RPE). Using immortalized RPE cells, serum or LPA stimulation is shown to activate the PI3K pathway, and its inhibition reduces cilia disappearance after adding these stimuli. PI3K lipid products are shown to be present in the transition zone of primary cilia and to be synthesized by PI3K β in RPE cells, but in murine embryonic fibroblasts, EGF stimulation increases PIP₃ through both PI3K α and β . Constitutively active PI3K α is shown to induce cilia disassembly in RPE cells, as does pharmacologic activation of the enzyme. Loss of PI3K activity, conversely, increases the proportion of ciliated cells in two lines (models for prostate hyperplasia and lung cancer). PI3K α activation leads to phosphorylation of S466 in centriolar protein CEP170, and phosphomimetic mutants of this protein similarly promote cilia disassembly in HEK293 cells. Finally, using knockdown or pharmacologic inhibition, requirements for PDK1, PKC δ and CEP170 are shown to be required for PI3K α -induced cilia disassembly.

Overall, the manuscript is clearly written, and the experiments produced robust data that are clearly presented. There are two points of novelty here: firstly, a mechanism whereby PI3K activation can lead to cilia-loss associated phenotypes, both in PROS and possibly in cancer or neoplasms. Secondly, a potentially novel signaling axis via PKC δ to CEP170. These are conceptual advances in the fields of cilia biology and disease as well as in signal transduction and PI3K signaling. Therefore, the manuscript justifies publication in a wide focus and broadly disseminated journal like Nat. Comms. Overall, I was convinced by the data and believe the story is largely complete.

That said, I did note some specific areas where the manuscript could be strengthened:

Authors' comment: We are very pleased with the very positive comments from Reviewer 1 and have taken action to address the specific areas highlighted, by performing additional experiments to strengthen the PIP₃ data and by textual changes to provide additional information or clarification.

(1) **Requirements of PI3K for cilia disassembly:** treatment of cells with serum seems to be the most robust stimulus for depleting cilia and yet, from the data presented in figures 3 and 7, blockade of the pathway does not completely block cilia disassembly in all cells.

Notably, the statistical comparison of inhibitor treatment to starved cell control is not presented, but there does still appear to be a difference even in GDC-0941 treated cells.

This is addressed in the manuscript with the statement that *"PI3K and downstream signalling is only one, but a physiologically critical, component of the serum-induced cilia disassembly network"*. The reviewer agrees that these incomplete but nonetheless significant effects could disrupt cilia-associated physiology to a phenotype-inducing degree. However, these subtleties should be more clearly addressed in the discussion.

We agree with this last statement and interpretation of these data by this Referee and have now included a more detailed dialogue of these subtleties beyond the sentence in italics above in the revised discussion section (page 10, line 509-513).

With regards to the comment on the statistical comparisons between starved cells and serum-stimulated + inhibitor treated cells, these comparisons were not included in Fig. 3 and Fig. 7 of the original manuscript as we felt there would have been too many lines/stars on the graph to interpret and appreciate the main message of these figures. Below, we have (1) reproduced Fig. 7h to demonstrate how complex the current presentation of this figure already is, and (2) included a table with the full post-hoc analysis of this dataset. In the end, we have decided not to include the additional comparisons in the figure itself as this is likely to make the figure uninterpretable.

Tukey's multiple comparisons	Mean Diff.	95.00% CI of diff.	Below threshold?	Summary	Adjusted P Value
starve vs. DMSO	34.40	23.71 to 45.09	Yes	****	<0.0001
starve vs. BYL719	22.40	11.71 to 33.09	Yes	****	<0.0001
starve vs. TGX-221	21.00	10.31 to 31.69	Yes	****	<0.0001
starve vs. GDC0941	18.40	7.713 to 29.09	Yes	****	<0.0001
starve vs. GSK2334470	10.60	-0.08723 to 21.29	No	ns	0.0532
starve vs. MK2206	16.40	5.713 to 27.09	Yes	****	0.0005
starve vs. 229	16.00	5.313 to 26.69	Yes	****	0.0007
DMSO vs. BYL719	-12.00	-22.69 to -1.313	Yes	*	0.0191
DMSO vs. TGX-221	-13.40	-24.09 to -2.713	Yes	**	0.0063
DMSO vs. GDC0941	-16.00	-26.69 to -5.313	Yes	***	0.0007
DMSO vs. GSK2334470	-23.80	-34.49 to -13.11	Yes	****	<0.0001
DMSO vs. MK2206	-18.00	-28.69 to -7.313	Yes	****	0.0001
DMSO vs. 229	-18.40	-29.09 to -7.713	Yes	****	<0.0001
BYL719 vs. TGX-221	-1.400	-12.09 to 9.287	No	ns	0.9999
BYL719 vs. GDC0941	-4.000	-14.69 to 6.697	No	ns	0.9223
BYL719 vs. GSK2334470	-11.80	-22.49 to -1.113	Yes	*	0.0222
BYL719 vs. MK2206	-5.000	-16.69 to 4.687	No	ns	0.5127
BYL719 vs. 229	-6.400	-17.09 to 4.287	No	ns	0.5354
TGX-221 vs. GDC0941	-2.600	-13.29 to 8.087	No	ns	0.9926
TGX-221 vs. GSK2334470	-10.40	-21.09 to 0.2872	No	ns	0.0611
TGX-221 vs. MK2206	-4.600	-15.29 to 6.087	No	ns	0.8532
TGX-221 vs. 229	-5.000	-15.69 to 5.687	No	ns	0.7936
GDC0941 vs. GSK2334470	-7.800	-18.49 to 2.887	No	ns	0.2925
GDC0941 vs. MK2206	-2.000	-12.69 to 8.697	No	ns	0.9985
GDC0941 vs. 229	-2.400	-13.09 to 8.287	No	ns	0.9954
GSK2334470 vs. MK2206	5.800	-4.887 to 16.49	No	ns	0.8511
GSK2334470 vs. 229	5.400	-5.287 to 16.09	No	ns	0.7254
MK2206 vs. 229	-0.4000	-11.09 to 10.29	No	ns	>0.9999

Fig. 7h: Reproduced from original manuscript.

Table 1: Tukey's post-hoc analysis of Fig. 7h.

(2) Localization of the PI3K-induced signal leading to cilia disassembly: Given the identification of CEP as a downstream effector of the pathway, and the long-established roles of phosphoinositides up- and down-stream of PI3K at primary cilia, the presence of PIP₃ at the transition zone reported in figure 4 makes sense. The dependence solely on PI3Kbeta for a resting pool is surprising but convincing. Here, EGF induced increases in ciliary PIP₃ are convincingly shown to require both PI3K alpha and beta in embryonic fibroblasts, which seem to have been selected due to the ability to use Pik3ca null lines.

However, the majority of the manuscript explores the effects of serum (and to a lesser extent, LPA) on cilia disassembly in RPE1 cells. Therefore, the manuscript would be substantially strengthened if effects of 1938, or serum ± PI3K inhibitors on ciliary PIP₃ were shown in RPE-1 cells. This would unambiguously assign this pool of PIP₃ as the pool that activates the cilia disassembly pathway.

We have included new data in the revised manuscript examining the effect of 1938 stimulation on ciliary PIP₃ levels in hTERT-RPE1 cells (Fig. 4e). To this end, hTERT-RPE1 cells were serum-starved for 48 hours and stimulated with 1938 for 5 minutes, followed by fixation and immunostaining with PIP₃ and ARL13B antibodies. We observed an increase in ciliary PIP₃ signals following 1938 treatment (Fig. 4e). This observation is consistent with the data included in the initial submission which showed 1938 also increased ciliary PIP₃ in MEFs (Fig. 4d). However, the response was more subtle in the hTERT-RPE1 cells compared to MEFs, likely because the basal unstimulated ciliary PIP₃ levels are higher in hTERT-RPE1 cells thereby reducing the dynamic range for this experiment in this cell line.

(3) Robustness of the PIP₃ localization data shown in figure 4: comparing the control conditions between RPE cells in 4a and 4c, the cells in figure 4c seem to have less PIP₃ than the cells in figure 4a, perhaps even as low as cells in figure 4a treated with TGX-211. This undermines the robustness of the results somewhat.

The cells in Fig. 4a and 4c are from different experiments which were stained at different times and imaged on different days, with different laser settings and the brightness adjusted separately in Fiji(ImageJ).

In Fig. 4a we are studying a control compared to a treatment that *decreased* the PIP₃ fluorescence intensity. In contrast, Fig. 4c studies a control compared to a treatment that *increased* the PIP₃ intensity.

Therefore, to maximise the dynamic range for each experiment, the laser intensity/gain settings and the brightness of the images for each experiment were adjusted distinctly to ensure the brightest condition within that experiment (the control of Fig. 4a, and the experimental for Fig. 4c) was not overexposed and then this was applied to all conditions within that experiment.

It is therefore it is not appropriate to directly compare the fluorescence intensity of cells in Fig. 4a and Fig. 4c, and no conclusions can be drawn from performing such a comparison as they reflect only technical differences in the experiment, masking any potential biological differences.

In reply to this Referee comment, and to avoid any potential confusion, we have added the following comment in the legend to Fig. 4 to highlight this to the reader (page 21, line 1036-1038, '*For all phosphoinositide imaging experiments using different treatments, the laser intensity, gain and brightness were adjusted independently to optimise the dynamic range of the experiment and applied to all conditions within the experiment*').

The data might be strengthened by plotting them differently, perhaps as a “super-plot” (doi: 10.1083/jcb.202001064) indicating the value of each cell in a scatter plot with experimental means indicated and the individual experiments color coded. If the experimental replicates are overlapping in each case and the difference between 4a and 4c is due to some other difference in experimental protocol, e.g. transfection vs not, this should be apparent. This will also eliminate any skew from an outlier experiment.

In response to the Reviewer’s suggestion, we have plotted the data from Fig. 4a and Fig. 4c as SuperPlots below:

Response to Reviewers’ comments - Fig. 1: PIP₃ MFI data from original manuscript Fig. 4a and Fig. 4c presented as SuperPlots. Each colour represents data as an independent experiment. The small points indicate the MFI of the ciliary PIP₃ signal in an individual cell and the large points indicate the mean MFI of the ciliary PIP₃ signal for each independent experiment.

We are very pleased to note that presenting the data in this way confirms that although there is some random variability between experiments, there is no outlier experiment that skews the trend.

Although it is not correct to directly compare the ciliary PIP₃ fluorescence intensity of cells in Fig. 4a with that of cells in Fig. 4c, due to the use of independent laser intensity and gain setting (as explained under (3) above), if one was to make this comparison it is observed that the mean PIP₃ MFI for the DMSO control in Fig. 4a is ~1600 arbitrary units, whereas the mean PIP₃ MFI for the vector control in Fig. 4c is ~2100 arbitrary units (Response to Reviewers’ comments - Fig. 1). This argues against the Reviewer’s postulate that ‘*cells in Fig. 4c seem to have less PIP₃ than the cells in Fig. 4a*’, further confirming that the images in Fig. 4a should not be directly compared with images in Fig. 4c.

We agree with the Reviewer that SuperPlots are an informative tool for cell biology data allowing the presentation of the maximum amount of information and to compare technical variability between

independent experiments. However, this graph style relies on mean and SD as measures of data centre and spread. The data presented in Fig. 4 of the manuscript do not uniformly fit the normal distribution (as confirmed using D'Agostino & Pearson test for normal distribution and can be appreciated from both the histograms (Fig. 4a,c) and SuperPlots (Response to Reviewers' comments - Fig. 1)). Non-normally distributed data are not well represented using summary statistics such as the mean and SD. Therefore, the SuperPlot is not the most suitable presentation style for this dataset. We have opted to continue to use histograms to present data in Fig. 4 as this graphical style more adequately represents non-normally distributed data and shows the variability at the single cell level.

Technical points:

(4) Extended figures 2 and 5 are posted at a resolution that is too low to really examine and interpret the data.

We apologise that the resolution of these figures was reduced during the file format conversion for manuscript submission. The resolution of these figures has now been increased and for submission of the revised manuscript these files have been exported in a format that does not compromise the resolution.

(5) Figure 3g left and right panels. This panel is a somewhat tricky game of spot the difference. The reader might benefit from a more accentuated highlighting of the 24 h serum vs 48 h serum in the two panels.

We thank the Reviewer bringing our attention to this potential confusion and the opportunity to improve clarity for the reader. Following this comment, we realised that while both panels of Fig. 3g were labelled correctly on the x-axis, the right panel was incorrectly labelled at the top, for which we apologise. In the revised manuscript this label has been corrected and we have highlighted the difference between the Fig. 3g left panel (24 h serum treatment) and the right panel (48 h serum treatment) by highlighting the timepoints in bold above the graphs.

(6) Many figure panels rely on scoring the % ciliated cells. The data are correctly reported as n=1 per experiment. However, the reviewer did not find an indication of how many cells were scored per experiment. This would be beneficial for assessing the robustness of the score.

In the revised manuscript, for all figures reporting on % ciliated cells we have included a statement of the number of cells scored per condition and per independent experiment in the corresponding figure legends. For most *in vitro* assays in hTERT-RPE1 cells, 100 cells were scored per condition per independent experiment. For assessment of ciliated endothelial cells in the murine embryonic retina, at least 397 cells were scored per mouse for n=3 mice per genotype.

Reviewer #2:

Authors' comment: We thank Reviewer 2 for constructive feedback and comments on our manuscript, all of which we can confidently address by providing more detailed explanations/clarification and performing additional experiments, as detailed below.

This manuscript described a transgenic mouse line carrying *Pik3ca*-H1047R/+ that showed embryonic lethality at E9.5 with some phenotypes like some of ciliopathy phenotypes. The authors suggested that these developmental abnormalities were caused by PI3Ka/b-induced cilia instability. It is appreciated that unbiased proteomics analyses were performed to study the global changes following hyperactivation of PI3Ka. However, PI3Ka and PI3Kb are both ubiquitously expressed in cell and activating either of them triggers numerous downstream signaling events including the ones that facilitate cell proliferation and indirectly promote cilia disassembly. The lack of clear mechanistic evidence demonstrating the direct, cilium-specific role of PI3Ka/b severely dampened the significance of their findings.

The ciliopathy field has long held a centriole/cilium-centric view, that ciliopathies are caused by mutations in proteins *specifically/selectively* localised and functioning at the centriole or cilium. However, over recent years it has become increasingly apparent that this is only the case in a subset

of ciliopathies, known as 'first-order'-ciliopathies¹. Moreover, many proteins initially thought to be cilia-specific have turned out to have additional non-ciliary localisations and functions, with some of the 'non-ciliary' ciliopathy proteins indeed localising to cilia, albeit transiently or in specific cell types². The historic narrow lens of 'first-order' ciliopathies is now thought to have hindered progress in the area and have limited the discovery of cilia contributions to disease.

Here, we focus on PI3K α and PI3K β , which are ubiquitously expressed proteins that regulate numerous downstream effector pathways, including proliferation³, as Reviewer 2 highlights. We appreciate Reviewer 2's concern of the possibility that PI3K-induced cilia phenotypes could be an indirect consequence of an effect on well-established PI3K functions such as proliferation, and we initially shared this concern. However, using multiple assays in several cell lines we instead found that under the cellular conditions in which PI3K α and PI3K β impact ciliary phenotypes, there is no overt perturbation of cell proliferation (Supplementary Fig. 4). Furthermore, we provide clear mechanistic evidence by mapping a signalling pathway downstream of PIP₃ leading to cilia disassembly. Multiple components within this pathway including PIP₃, PDK1, PKC ι , CEP170 and KIF2A do localise to the cilia base or centrosome. We also show that inhibition or knockdown of these pathway components fully rescues the cilia instability induced by activated PI3K α signalling, showing specificity to this cilia-associated pathway, rather than an indirect effect of other PI3K α effector proteins. We hypothesise that PI3K α and PI3K β also localise to the primary cilium either constitutively or transiently but unfortunately the field lacks the tools to address this interesting question.

Collectively, our data and current and emerging concepts in the primary cilia field support our hypothesis that PI3K α may act like many well-established ciliopathy proteins, whereby it functions both at the cilium and outside the cilium, has functional effects both on cilia assembly/structure/function and other cell biological processes, with its dysfunction causing a phenotype that involves (but is not limited to) a cilia contribution.

In summary, we do not view the fact that PI3K α has non-ciliary functions as an issue or limitation of the data presented here, and fully acknowledge its other functions in the manuscript. We thank Reviewer 2 for raising this important comment and have therefore clarified this recently described concept of cilia field in the revised manuscript (page 2, line 63-66).

1. PI3K-H1047R somatic mutation in human causes significant overgrowth, leading to hemihypertrophy and malformation of the vascular system. Polydactyly was observed in CLOVE patients but rarely. No progressive ciliopathy phenotypes such as progressive polycystic kidneys is observed in patients. It is far-fetched to conclude that the pathogenic mechanism of PI3Ka-H1047R is that it causes primary cilia instability.

We agree with the Reviewer that not all *PIK3CA*^{H1047R} phenotypes are likely to be caused by primary cilia instability. Instead, we speculate that *PIK3CA*^{H1047R}-related diseases may be disorders with cilia contribution, whereby cilia instability is a contributing factor to a subset of the phenotypes but not the whole pathogenic mechanism. We propose that this is the case in terms of the polydactyly observed in PROS⁴. Significantly, we have found additional evidence in the literature that renal cysts are also observed in PROS-affected individuals^{4, 5}, a condition which can also be associated with primary cilia dysfunction. We have now included this additional clinical evidence of renal cysts in PROS in the revised manuscript to strengthen the contention that aspects of the PROS phenotype may be linked to cilia instability (Page 9, line 464-467 Page 9, line 469-470).

Disorders with ciliary contributions are a recently described group which is estimated to consist of >300 diseases not classified as ciliopathies but where cilia dysfunction contributes to the phenotype, largely by unknown mechanisms¹.

We acknowledge that polydactyly and likely renal cysts are rare phenotypes in PROS, however, posit that just because they are uncommon does not necessary mean they are unconnected, for several reasons. Firstly, PROS is a mosaic condition with the variability of phenotypes largely driven by the differential anatomical distributions of mutations⁶. Second, PROS is a rare disease which was only recognised a decade ago with a limited number of subjects investigated, with the phenotypic spectrum continuing to emerge. Finally, diagnostic tests are generally performed on accessible regions of the

body that exhibit overgrowth⁶, and it is conceivable that reported phenotypes are biased towards more external features and that mild renal cysts could be missed. Taken together, these observations collectively suggest that it would be unwise to dismiss a phenotype just because it is rare in PROS. We suggest that our data might in fact help to further diagnose phenotypes in PROS subjects and have added a comment to the discussion as follows: '*We suggest that our current report may help to further diagnose this type of a cilia-related phenotypes in PROS-affected individuals*' (Page 9, line 472-474).

Throughout the manuscript, we have been extremely careful to highlight that the potential classification of PROS as a disorder with ciliary contribution is a speculation and have ensured that this caution is adequately conveyed in the abstract and the revised manuscript (page 9, line 468-471, Page 11, line 556-557). Given the rare nature of PROS and its only recent recognition with clinical evidence still emerging, our data and speculation may stimulate further clinical investigation on cilia-related phenotypic features.

2. INPP5E is specifically localized in primary cilia. Inhibiting INPP5E also causes the increase of PIP3 in the transition zone area and decreases the stability of primary cilia. However, the INPP5E-null mice don't share similar phenotypes as the PIK3ca H1047R/+ mice. MORMS (mutated INPP5E not localizing to primary cilia) and JBTS1 (mutated INPP5E with less phosphatase activity) patients, also don't show similar phenotype as patients expressing PI3K-H1047R.

This is an interesting point raised by the Reviewer. There are several reasons to explain why *PIK3CA*^{H1047R} mutant mouse embryos or humans do not resemble mouse embryos or human subjects with *INPP5E* deletion or mutations, as detailed below. We recognise that these subtleties may not be clear to the general reader but, given our own extensive studies and knowledge of *INPP5E* (S.E.C.), we are expertly positioned to comment on this subject.

In reply to Referee 2, we have now provided a more detailed, nuanced and extensive discussion of the differences between *INPP5E* and *PIK3CA* models in the discussion of the revised manuscript (Page 11, lines 546-555), which summarizes the points made below.

- (1) We and others have previously reported that although deletion of *Inpp5e* increases ciliary PIP₃, *Inpp5e* null cells do not exhibit reduced cilia stability unless they are treated with serum or express an oncogene^{7, 8, 9, 10}. We reported this phenomenon previously as a threshold effect, whereby it is not only increased ciliary PIP₃ but an increase in PIP₃ above a threshold that is required to induce cilia disassembly⁸. In contrast *PIK3CA*^{H1047R} expression or 1938 treatment alone are sufficient to reduce cilia stability. In this context, it is important to note that *INPP5E* loss on its own does not induce signalling but instead results in an incapacity to suppress signalling by other stimuli. This contrasts with *PIK3CA* and 1938 which induce signalling on their own. We argue here that unlike *Inpp5e* deletion alone, PI3K hyperactivation is sufficient to overcome the ciliary PIP₃ threshold mentioned above. Therefore, in conclusion, cilia remain present in *Inpp5e*-null embryos^{7, 9} whereas cilia instability likely underlies the cilia-related phenotypes in *PIK3CA*^{H1047R} embryos.
- (2) The phenotype of *Inpp5e*-null embryos is not driven by reduced cilia stability but is caused by mislocalisation of signalling components to cilia^{7, 11, 12}. It is possible that *PIK3CA*^{H1047R} expression also perturbs the localisation of components to cilia. However, given that cilia loss is likely to have more severe phenotypic consequences than mislocalisation of components in the subset of cells that retain this organelle, we focused on the mechanisms of reduced cilia stability in this manuscript.
- (3) It is not possible to examine *PIK3CA*^{H1047R} embryos for specific *Inpp5e*-null phenotypes such as polydactyly, cystic kidneys, anophthalmia reduced ossification and pulmonary hypoplasia^{7, 9} as these affected organs develop *after* the developmental timepoint of *PIK3CA*^{H1047R} embryonic lethality.
- (4) Conversely, the early embryonic lethality and more severe phenotype of *PIK3CA*^{H1047R} embryos is likely not observed in *Inpp5e*-null embryos because (1) loss of cilia caused by reduced cilia stability is likely to induce a more severe phenotype compared to mislocalisation of components to the cilia which still form, and (2) *PIK3CA*^{H1047R} expression also perturbs other pathways

- critical at this time of development which contribute to the phenotype, including angiogenesis¹³.
- (5) In humans it is difficult to directly compare MORM/Joubert syndrome-affected individuals with PROS-affected individuals given that MORM/Joubert syndrome results in a ubiquitous partial loss of function of *INPP5E*^{9, 14} (which is unlikely to exceed the PIP₃ threshold required to induced cilia instability), whereas PROS results from a mosaic constitutive activation of PI3K α ⁶ (which may overcome the PIP₃ threshold for cilia instability, but in only some cells).

3. It is interesting that PIP3 level in the transition zone changed when PI3Ka/b was activated or inhibited. However, no studies were conducted to analyze the direct consequence of this PIP3 change in the context of primary cilia. For example, was INPP5E in cilia or the PI4P-PI4,5P2 distribution along the cilium changed? What about other structural, trafficking, and signaling molecules residing/functioning in cilia?

We respectfully disagree with the contention that we have not analysed the direct consequences of altered PIP₃ on primary cilia. We argue that the study of primary cilia assembly/disassembly dynamics is analysis of a direct consequence of ciliary PIP₃ levels on primary cilia biology. This covers Fig. 3, 5, 7 in our manuscript.

In terms of the suggestion to examine the consequences of altered ciliary PIP₃ on **localisation of INPP5E** and associated PI(4)/PI(4,5)P₂, we posit that there is no rationale for such an experiment. Perhaps the Reviewer is referring to the Phau et al 2017¹⁵ publication, which showed that INPP5E moves out of the cilia upon serum stimulation and therefore hypothesizes that PI3K activation/increased ciliary PIP₃ will phenocopy serum stimulation to induce INPP5E ciliary exit. However, using live cell imaging we previously showed that INPP5E constitutively localises to the ciliary transition zone regardless of serum stimulation¹⁶, and that this possible speculation of this Referee is incorrect. We therefore contend that there is no rationale to expect modulation of ciliary PIP₃ levels to alter the ciliary localisation of INPP5E and associated PI(4)/PI(4,5)P₂.

However, regarding the suggestion of Referee 2 to examine the consequences of altered ciliary PIP₃ on **localisation of other signalling molecules at the cilium**, we examined the localisation to cilia of the Smoothed (SMO) G protein-coupled receptor in *PIK3CA*^{H1047R}-expressing hTERT-RPE1 cells, given that we previously reported that *Inpp5e* deletion and increased ciliary PIP₃ is associated with reduced retention of this receptor at the cilium⁷. Ciliated vector control and *PIK3CA*^{H1047R} hTERT-RPE1 were stimulated +/- the SMO agonist SAG for 24 h, immunostained with SMO antibodies, imaged by confocal microscopy and the ciliary SMO mean fluorescence intensity measured. We observed a robust accumulation of SMO in the cilia axoneme in SAG-treated vector control cells, which was partially abrogated by *PIK3CA*^{H1047R} expression (Response to Reviewers' comments - Fig. 2). This observation is consistent with disrupted transition zone barrier function in cells with high ciliary PIP₃ levels. Reduced ciliary SMO would contribute to the repression of Hedgehog signalling observed in these cells, but the reduced cilia stability and therefore a reduced population of cells capable of transducing the signal is likely to have a more profound effect. We have included these new data in the revised manuscript (Supplementary Fig. 1g). Please note, to complete the initial characterisation of the *PIK3CA*^{H1047R} hTERT-RPE1 dataset in supplementary Fig. 1, we also moved the original Fig. 5a to supplementary Fig. 1e.

Response to Reviewers' comments - Fig. 2: Vector and $PIK3CA^{H1047R}$ -transduced hTERT-RPE1 cells were serum-starved for 48 h with 0.5 $\mu\text{g/ml}$ doxycycline +/- 24 h 200 nM SAG, fixed, stained with acetylated tubulin (red) and SMO (green) antibodies and DAPI and imaged by confocal microscopy, bar indicates 1 μm . The ciliary SMO MFI was measured and presented as a histogram. $n=90$ cells per condition from 3 independent experiments **** $p<0.0001$ (Kolmogorov-Smirnov test for $PIK3CA^{H1047R}$ + SAG vs Vector + SAG).

Moreover, checking the accumulation of Gli3 at the ciliary tip and decrease of ciliary GPR161 induced by SAG is much more relevant in this scenario than checking the global transcription level of Gli1.

We respectfully disagree that examining GLI3 and GPR161 localisation to cilia in response to SAG are more relevant readouts of Hedgehog signalling than assessment of the *Gli1* and *Ptch1* transcriptional responses in this case. Indeed, *Gli1* and *Ptch1* are considered 'gold-standard' target genes widely used for assessment of Hedgehog pathway activation in mammalian cells¹⁷. Cells and embryos expressing $PIK3CA^{H1047R}$ exhibit a reduced proportion of ciliated cells that can respond to the Hedgehog signal (or SAG). It is clear that it is not possible (or relevant) to examine the localisation of components (such as GLI3 or GPR161) to cilia in cells that have no cilium.

As *Gli1* is a Hedgehog target gene, assessment of *Gli1* transcription provides information on the ability of the cell population to respond to Hedgehog signals (or SAG) and captures both the remaining ciliated population as well as the cells that have lost their cilium. In contrast, examining GLI3 or GPR161 localisation at cilia would only assess the small cell population retaining their cilium and would not take into account the major PI3K-induced phenotype of cilia instability.

Subsequently, if this increase of PIP₃ is diminished/blocked specifically at the transition zone, would any ciliary changes (including the increased cilia instability) be inhibited?

We interpret this question as asking whether specifically targeting a phosphatase to the cilia transition zone to deplete the excess PIP₃ produced by PI3K would rescue the phenotypes. Logical candidates for this type of experiment would be the phosphatases INPP5E and PTEN. However, both candidates present a number of issues which would prohibit the interpretation of such an experiment:

In terms of INPP5E, indeed this phosphatase localises to the primary cilia and when imaged live or with carefully optimized immunofluorescence conditions localises to the transition zone¹⁶. However, INPP5E is a 5-phosphatase which hydrolyses PIP₃ to produce PI(3,4)P₂ (and not PI(4,5)P₂, the substrate used by PI3K used for PIP₃ production) which also recruits and activates effector proteins with overlapping specificity as PIP₃¹⁸. Furthermore, INPP5E also dephosphorylates PI(4,5)P₂¹⁹, which localises in very close proximity to PIP₃ at the inner transition zone membrane¹⁶ and therefore INPP5E overexpression at the transition zone will disrupt both phosphoinositide lipids and their downstream effector pathways.

PTEN is the other candidate phosphatase. PTEN dephosphorylates PIP₃ to produce PI(4,5)P₂ thereby directly opposing PI3K signalling. However, targeting PTEN to cilia would be technically difficult and any data difficult to interpret, for the reasons below:

- 1) PTEN has not been shown to naturally localise to the cilia transition zone and would need to be targeted by an engineered motif to the transition zone.
- 2) To the best of our knowledge, even though there are multiple well-established ciliary axoneme targeting signals, no specific transition zone targeting motifs have been defined.
- 3) PTEN localisation has been shown to be differentially aberrantly affected by N- and C-terminal epitope tagging and different tags (i.e. HA vs GST vs Flag) have different effects on localisation^{20, 21}.

4. Another very interesting and important question derived from the observation of PIP₃ changes at TZ in response to PI3Ka/b activity: if PI3Ka/b is not specifically activated at the TZ, how could PIP₃ at the TZ be changed? What is the intermediate player(s) transferring PI3K activity/product from the cytosol/plasma membrane to the TZ?

We agree with the Reviewer that the localisation of PI3K isoforms to the cilia transition zone or the mechanism of phosphoinositide recruitment to the transition zone if these lipids are not produced there is interesting.

Using superresolution microscopy, we have previously shown that the localisation of PIP₃ at the transition zone is consistent with the inner leaflet of the transition zone membrane¹⁶. PI(4,5)P₂, the substrate for PI3K-mediated PIP₃ production, also localises to the inner leaflet of the transition zone membrane, albeit in a slightly different axial subdomain¹⁶. Therefore, the most logical hypothesis is that a pool of PI3K α and PI3K β constitutively or transiently localise to the transition zone where they phosphorylate the PI(4,5)P₂ *in situ* to produce PIP₃ directly on the inner leaflet of the transition zone membrane when stimulated by upstream growth factor receptors. We foresee no direct arguments against this hypothesis, other than the lack of direct evidence to date.

Very unfortunately, there is a lack of sensitive and specific PI3K α or PI3K β antibodies for immunofluorescence. Furthermore N-terminal tagging these enzymes aberrantly stimulates kinase activity while C-terminal tags render them kinase-dead. Therefore, assessment of the subcellular localisation of PI3K has remained impossible in the field. We briefly discussed this question in the original manuscript and in reply to this Referee comment, in the revised discussion we have elaborated on the specifics of this hypothesis and current technical limitations (Page 10, line 494-502).

5. The observation that CEP170 can be phosphorylated at S466 provides an interesting lead for mechanistic analysis, which is lacking in the current manuscript. The HEK293 cell line is not a good model to study primary cilia.

We agree that HEK293 cells are not an ideal model to study primary cilia, as we had acknowledged in the original manuscript (*Page 7, line 366-369 'Although the proportion of serum-starved HEK293 cells that form cilia is relatively low, this cell line has been used previously to study primary cilia²³ and was chosen here given its relative ease of transfection and the fact that we were unable to express HA-CEP170 plasmids in hTERT-RPE1 cells, likely due to the large DNA insert size.'*). However, as we had referenced, the HEK293 cell line has been used previously to study primary cilia biology²³. Given the limitations of HEK293 in the context of primary cilia, we restricted our use of these cells to a single experiment of our manuscript, which for technical reasons could only be performed in this cell model as explained.

We would ideally like to perform the CEP170 phosphomimetic (HA-CEP170(S466D)) analysis on cilia in hTERT-RPE1 cells and we attempted this experiment in hTERT-RPE1 cells. However, our efforts to express HA-CEP170 and HA-CEP170(S466D) plasmids in these cells were unsuccessful as we were unable to detect expression of the recombinant proteins (Response to Reviewers' comments - Fig. 3).

This is most likely due to the insert size of these plasmids being very large (4755 nucleotides), with large plasmids size known to significantly limit transfection efficiency in most cell types.

Response to Reviewers' comments - Fig. 3: *hTERT-RPE1* cells were transfected with HA vector, HA-CEP170 or HA-CEP170(S466D) and immunoblotted with antibodies to HA or Vinculin. Cells expressing a ~120 kDa HA-tagged protein were used as a positive control.

In contrast, HEK293 cells are more easily transfected, even with large plasmids, and we were able to successfully express the HA-CEP170 plasmids in this cell line (Response to Reviewers' comments - Fig. 4). Therefore, HEK293 cells are the only viable cell line model for this experiment.

Response to Reviewers' comments - Fig. 4: HEK293 cells were transfected with HA vector, HA-CEP170 or HA-CEP170(S466D) and immunoblotted with antibodies to HA or CEP170.

6. Whether/how the AURKA signaling pathway is influenced by modifying PI3K α /b activity and the subsequent influence on cilia stability should be determined.

AURKA is a well-established inducer of primary cilia disassembly. AURKA activity is regulated at multiple levels which impacts on its ability to induce cilia disassembly, including via activating phosphorylation at T288²⁴ and regulation at the transcriptional level^{10, 25}. In response to this Referee comment, we have now interrogated our phosphoproteomics datasets and performed additional experiments, but observed no evidence that AURKA activity is modulated by PI3K α or PI3K β .

Firstly, total AURKA mRNA and proteins levels have been shown to be regulated by AKT/mTORC1 signalling in the context of *INPP5E* or *VHL* knockdown^{10, 25}. Chowdhury *et al.* 2021²⁵ showed that 48 h treatment of hTERT-RPE1 cells with the dual PI3K/mTOR inhibitor NVP-BE235 in serum-free media reduced AURKA protein levels compared to vehicle control treatment. As this was the most relevant experiment to our experimental question (without the complication of *INPP5E* or *VHL* knockdown), we performed a similar experiment by treating hTERT-RPE1 cells with the pan-class I PI3K inhibitor GDC0941 for 48 h in serum-free media followed immunoblotting using AURKA antibodies. We did not observe a robust difference in AURKA protein levels between GDC0941 and vehicle treated cells (Response to Reviewers' comments - Fig. 5).

Response to Reviewers' comments - Fig. 5: *hTERT-RPE1* cells were cultured in serum or serum-starved +/- 0.5 μ M GDC0941 for 48 h and immunoblotted with AURKA and GAPDH antibodies.

Secondly, AURKA is activated by phosphorylation of T288²⁴. We did not detect the pAURKA(T288) containing peptide in either of our phosphoproteomic datasets (Supplementary table 4, 5). A key AURKA substrate is HDAC6, which is phosphorylated at an undefined site upon AURKA activation²⁴. We also did not detect any HDAC6 phosphopeptides in our phosphoproteomic experiments (Supplementary table 4, 5). We acknowledge that in both cases this is an absence of evidence, rather than evidence that AURKA is inactive.

Therefore, we interrogated our kinase-substrate enrichment analysis (KSEA) of *hTERT-RPE1* cells treated with 1938 or insulin +/- BYL719 for modulation of the AURKA substrate group, as an indication of whether AURKA is activated by PI3K α under our experimental conditions. The AURKA substrate group was included in the Omni and PhosphSitePlus databases. We observed no significant changes in AURKA activity (Response to Reviewers' comments - Tables 2, 3). We used the phosphoproteomic studies and KSEA as an untargeted approach to prioritise candidate downstream of PI3K signalling effectors in disassembly to follow up.

In summary, given that AURKA was not identified as an activated kinase in these conditions, it was not prioritised for follow up. However, given its established role in cilia disassembly, links to PI3K signalling, and in response to this Referee comment, we have now added a comment to the results section of the revised manuscript to indicate that no changes in the AURKA activity were observed (page 8, line 424-426).

	p-value	adjusted p-value	fold enrichment (vs DMSO)
1938 15 min	0.169638874	0.604553213	2.201919893
1938 4h	0.219024189	0.790199509	0.746314253
1938 + BYL 15 min	0.945795163	0.959806943	1.016313917
1938 + BYL 4 h	0.24688376	0.977920335	1.232648892
Insulin 15 min	0.253050056	0.732208905	1.669466821
Insulin 4 h	0.343674047	0.949250683	1.216288479

Response to Reviewers' comments - Table 2: KSEA using the Omni database for the AURKA substrate group in 24 h serum-starved *hTERT-RPE1* cells stimulated for 15 min or 4 h with 1938 or Insulin +/- BYL719 treatment, normalised to DMSO.

	p-value	adjusted p-value	fold enrichment (vs DMSO)
1938 15 min	0.027424247	0.108290618	1.920074459
1938 4h	0.204005638	0.45841678	0.82739842
1938 + BYL 15 min	0.492733922	0.499969813	0.018214365
1938 + BYL 4 h	0.493996547	0.493996547	0.015048994
Insulin 15 min	0.139946112	0.387068697	1.080561482
Insulin 4 h	0.259643535	0.488885899	0.644444757

Response to Reviewers' comments - Table 3: KSEA using the PhosphoSitePlus database for the AURKA substrate group in 24 h serum-starved hTERT-RPE1 cells stimulated for 15 min or 4 h 1938 or Insulin +/- BYL719 treatment, normalised to DMSO.

7. Data presentation in Fig. 4 is confusing. What MFI does each of the red dots (PIP3 or PI(3,4)P2) in the images correlate to as shown in the corresponding histogram?

The PI MFI was measured for each cilium in a box of a standardised size at the base of the primary cilium demarked by ARL13B immunoreactivity and centred around the highest intensity PI pixel. We had detailed this information in the methods section of the original manuscript (page 14, line 711-718) as follows: 'Ciliary PI MFI was measured using Fiji ImageJ in hTERT-RPE1 cells and MFEs. Cells were immunostained with PI antibodies and ARL13B antibodies to define the ciliary axoneme and imaged by confocal microscopy at the same laser scanning intensity and Airyscan processing. A macro was developed to identify the cilia axoneme using the ARL13B channel, skeletonize the signal and define the location. Boxes of standardised size were then drawn at either ended of the axoneme centred around the highest intensity PI pixels. The PI MFI was measured in each box and the MFI for the highest intensity box, representing the cilia base, returned. The PI MFI was plotted as a histogram using R.'

We appreciate that this could have been unclear for a reader focusing on the results section and figure legends. In response to this Referee comment, and to further clarify, we have now added more detailed information in the figure legends themselves (Page 20-21, line 994-1038).

Would the difference still significant if comparing the mean MFI of all cells quantified in each experimental condition?

Using Fig. 4a as an exemplar we have tested whether comparing the mean PIP₃ MFI of all cells quantified in each experimental condition using a one-way ANOVA also shows a significant difference in MFI between cells treated with TGX-221 or GDC0941 relative to DMSO controls. Indeed, this analysis confirmed a significant reduction in mean PIP₃ MFI following TGX-221 or GDC0941 treatment (Response to Reviewers' comments - Fig. 6). No significant difference was observed upon BYL719 treatment (Response to Reviewers' comments - Fig. 6), consistent with our original analysis.

Response to Reviewers' comments - Fig. 6: PIP₃ MFI data from original manuscript Fig. 4a. Each cell quantified is represented as a point, bars indicate mean +/- SD. n>75 cells per condition from 3 independent experiments ****

$p < 0.0001$ relative to DMSO control (one-way ANOVA $p < 0.0001$).

We have decided not to change the statistical analysis of the data presented in Fig. 4 in the revised manuscript because these data do not uniformly fit the normal distribution (as confirmed using D'Agostino & Pearson test for normal distribution and as can be observed from the histograms presented in Fig. 4). Non-normally distributed data are not well compared using summary statistics such as the mean. Furthermore, a fundamental assumption of the one-way ANOVA analysis of difference in population means is that the data is normally distributed. Therefore, we suggest that the Kruskal-Wallis test, which compares the distributions and does not rely on the mean, is a better way to analyse this data set and we have continued to use this test in the revised manuscript.

Reviewer #3:

There are some comments on the technical side of phosphoproteomics.

We thank the Reviewer for their technical suggestions which we do not view to be a concern with the experiment design or data. We have provided additional information and statistical analysis in the revised manuscript to answer these suggestions, thereby enabling the reader to more easily interpret the phosphoproteomic data sets.

1. Phosphosite Screening Process

The manuscript currently does not elaborate on the selection criteria for the phosphosites analyzed in the study. It is imperative to understand the basis on which these were chosen for subsequent analyses. Could you please provide a more detailed description of the screening process, including any statistical cutoffs, conservation analysis, or functional relevance that guided your selection?

In the revised manuscript, we have now provided a more detailed description of the phosphosite selection process in Fig. 2b and Supplementary Fig. 5a and the respective figure legends. Statistical cutoffs of $p < 0.05$ and a 2-fold change were used for selection of differentially regulated phosphosites and substrate groups (for Kinase Substrate Enrichment Analysis). Functional relevance was also assessed by comparing differentially regulated phosphosites with proteins contained by the cilia databases Syscilia and CiliaCarta (Fig. 2b). In the volcano plots (Fig. 6a, Supplementary Fig. 2b and Supplementary Fig. 5d) we have also indicated the number of phosphosites that were significantly ($p < 0.05$ and a 2-fold change) upregulated or downregulated by each treatment relative to DMSO.

2. Coefficient of Variation (CV) Absence

The manuscript does not currently report the coefficient of variation (CV) for the experimental conditions. The CV is a crucial statistical measure that can help ascertain the reproducibility and reliability of the observed results. I recommend that the authors include the CV calculations for each condition or provide a rationale for its omission.

CVs are often used as a metric of quantification in proteomics. However, they are not good measures of reproducibility in biological experiments as processing methods will impact CV calculations and low CVs do not directly translate towards accuracy in quantification²⁶. We have employed the MSstats statistical framework that uses a linear mixed-effect model to specify our experimental design and summarise biological and technical variation at the level of feature and phosphosite in our case. This results in robust phosphosite intensity summaries without overfitting the data. Variance across five biological replicates in each experiment has been taken into account in the calculation of p-values which put the magnitude of biological change in relation to technical variance. A full set of metrics is reported in Supplementary tables 4 and 5. Additionally, Pearson correlation coefficients for biological replicates are reported in Supplementary fig. 2a and 5c.

3. Choice of DDA over DIA

The use of a data-dependent acquisition (DDA) approach, particularly with a long gradient, raises questions about the efficiency of the mass spectrometry run. Given that the 150-minute gradient was not fully utilized, could you discuss the reasons behind opting for DDA rather than

data-independent acquisition (DIA), which may offer higher data integrity and depth? It would be beneficial for readers to understand the specific advantages that DDA provided for your study's objectives, as well as any considerations given to the use of DIA.

The choice of the acquisition method was guided by robust benchmarking of the analysis pipeline that led to consistency between a series of studies related to this biological project. The aim of this study was not to develop or utilise a method with an improved set of performance metrics that would overcome another method. Instead, we selected a reproducible method with clearly established false discovery rate estimations in our hands. The phosphoproteome coverage was sufficient to test the hypotheses defined in this study.

The proteomic field is still in the process of benchmarking DIA analyses and reports keep emerging with continuously changing acquisition parameters as well as data evaluation metrics. DIA data analysis tools can significantly vary in their outputs between themselves, and iterations of software versions keep providing incrementally varied outputs in a small space of time. Given the DIA analysis is still a rapidly developing area of proteomics, we continue to evaluate the robustness of our DIA pipeline. In our view, it is not in the scope of this manuscript to discuss these considerations.

References

1. Lovera M, Lüders J. The ciliary impact of nonciliary gene mutations. *Trends Cell Biol* **31**, 876-887 (2021).
2. Hua K, Ferland RJ. Primary cilia proteins: ciliary and extraciliary sites and functions. *Cell Mol Life Sci* **75**, 1521-1540 (2018).
3. Bilanges B, Posor Y, Vanhaesebroeck B. PI3K isoforms in cell signalling and vesicle trafficking. *Nat Rev Mol Cell Biol* **20**, 515-534 (2019).
4. Mirzaa G, Graham JM, Jr., Keppler-Noreuil K. PIK3CA-Related Overgrowth Spectrum. In: *GeneReviews* (eds Adam MP, *et al.*). University of Washington, Seattle (2023).
5. Gazzin A, *et al.* Work-Up and Treatment Strategies for Individuals with PIK3CA-Related Disorders: A Consensus of Experts from the Scientific Committee of the Italian Macroductyly and PROS Association. *Genes (Basel)* **14**, (2023).
6. Madsen RR, Vanhaesebroeck B, Semple RK. Cancer-Associated PIK3CA Mutations in Overgrowth Disorders. *Trends Mol Med* **24**, 856-870 (2018).
7. Dyson JM, *et al.* INPP5E regulates phosphoinositide-dependent cilia transition zone function. *J Cell Biol* **216**, 247-263 (2017).
8. Conduit SE, *et al.* A compartmentalized phosphoinositide signaling axis at cilia is regulated by INPP5E to maintain cilia and promote Sonic Hedgehog medulloblastoma. *Oncogene* **36**, 5969-5984 (2017).
9. Jacoby M, *et al.* INPP5E mutations cause primary cilium signaling defects, ciliary instability and ciliopathies in human and mouse. *Nat Genet* **41**, 1027-1031 (2009).
10. Plotnikova OV, *et al.* INPP5E interacts with AURKA, linking phosphoinositide signaling to primary cilium stability. *J Cell Sci* **128**, 364-372 (2015).
11. Chávez M, Ena S, Van Sande J, de Kerchove d'Exaerde A, Schurmans S, Schiffmann SN. Modulation of Ciliary Phosphoinositide Content Regulates Trafficking and Sonic Hedgehog Signaling Output. *Dev Cell* **34**, 338-350 (2015).
12. Garcia-Gonzalo FR, *et al.* Phosphoinositides Regulate Ciliary Protein Trafficking to Modulate Hedgehog Signaling. *Dev Cell* **34**, 400-409 (2015).
13. Hare LM, *et al.* Heterozygous expression of the oncogenic *Pik3ca*(H1047R) mutation during murine development results in fatal embryonic and extraembryonic defects. *Dev Biol* **404**, 14-26 (2015).
14. Bielas SL, *et al.* Mutations in INPP5E, encoding inositol polyphosphate-5-phosphatase E, link phosphatidyl inositol signaling to the ciliopathies. *Nat Genet* **41**, 1032-1036 (2009).
15. Phua SC, *et al.* Dynamic Remodeling of Membrane Composition Drives Cell Cycle through Primary Cilia Excision. *Cell* **168**, 264-279.e215 (2017).

16. Conduit SE, Davies EM, Fulcher AJ, Oorschot V, Mitchell CA. Superresolution Microscopy Reveals Distinct Phosphoinositide Subdomains Within the Cilia Transition Zone. *Front Cell Dev Biol* **9**, 634649 (2021).
17. Chen JK, Taipale J, Young KE, Maiti T, Beachy PA. Small molecule modulation of Smoothed activity. *Proc Natl Acad Sci U S A* **99**, 14071-14076 (2002).
18. Eramo MJ, Mitchell CA. Regulation of PtdIns(3,4,5)P₃/Akt signalling by inositol polyphosphate 5-phosphatases. *Biochem Soc Trans* **44**, 240-252 (2016).
19. Kisseleva MV, Wilson MP, Majerus PW. The isolation and characterization of a cDNA encoding phospholipid-specific inositol polyphosphate 5-phosphatase. *J Biol Chem* **275**, 20110-20116 (2000).
20. Gil A, *et al.* Nuclear localization of PTEN by a Ran-dependent mechanism enhances apoptosis: Involvement of an N-terminal nuclear localization domain and multiple nuclear exclusion motifs. *Mol Biol Cell* **17**, 4002-4013 (2006).
21. Mingo J, *et al.* A pathogenic role for germline PTEN variants which accumulate into the nucleus. *Eur J Hum Genet* **26**, 1180-1187 (2018).
22. Shnitsar I, *et al.* PTEN regulates cilia through Dishevelled. *Nat Commun* **6**, 8388 (2015).
23. Gerdes JM, *et al.* Disruption of the basal body compromises proteasomal function and perturbs intracellular Wnt response. *Nat Genet* **39**, 1350-1360 (2007).
24. Pugacheva EN, Jablonski SA, Hartman TR, Henske EP, Golemis EA. HEF1-dependent Aurora A activation induces disassembly of the primary cilium. *Cell* **129**, 1351-1363 (2007).
25. Chowdhury P, *et al.* Therapeutically actionable signaling node to rescue AURKA driven loss of primary cilia in VHL-deficient cells. *Sci Rep* **11**, 10461 (2021).
26. Ivanov MV, Garibova LA, Postoenko VI, Levitsky LI, Gorshkov MV. On the excessive use of coefficient of variation as a metric of quantitation quality in proteomics. *Proteomics* **24**, e2300090 (2024).

REVIEWER COMMENTS

Reviewer #1 (Remarks to the Author):

Overall, the authors have comprehensively responded to my concerns and there are no major issues remaining that would preclude publication in my opinion. I do have a couple of follow up points to consider in terms of data presentation which may still improve the manuscript:

(1) The reviewer appreciated the inclusion of the full stats table in the reviewer response. I believe this would be beneficial to include in the supplement, since it fully supports the conclusions (though I concede the authors' point that the comparisons are too numerous for inclusion in the figure).

(3) The reviewer appreciated the author's plotting the data from figures 4a and c as a super plot, which does alleviate the concerns about the robustness of the data. I would recommend using these super-plots in place of the histograms as they are much more informative. The author's are correct that showing the means and SD is not appropriate for non-Gaussian distributions, but there is no requirement to use mean and SD for super plots. Indeed, a statistical analysis was performed using Kruskal-Wallis, which compares medians. The super-plot could easily compare medians of each experiment and accomplish the same effect. Although showing mean \pm SD of medians would be ok (the variance between experiment medians is likely more gaussian), medians and 95% CI of the median could also be plotted.

Reviewer #2 (Remarks to the Author):

This reviewer is pleased by the authors' efforts on putting together a very organized, detailed, and clear responses to reviewers' comments. Here is the remaining concern, which I hope can be addressed better by carefully constructing the introduction and discussion.

The authors' argument about the 'first-order ciliopathies' and 'second-order ciliopathies' is confusing. In the review article from Lovera et al., 'first-order ciliopathies' refer to diseases caused by gene mutations directly affect the assembly, maintenance, or function of centrioles or cilia; and 'second-order ciliopathies', also termed as 'disorders with ciliary contribution', are caused by genes with indirect effects on cilia. Do the authors conclude their findings with PI3Ka/b direct effects or indirect effects on cilia? This needs to be clarified. To a broader interest, whether it's appropriate to call cancers 'disorders with ciliary contribution' would be a question remain to be discussed.

I agree with the authors that many "cilium-specific" proteins are later found with non-ciliary localization and function and it is true vice versa. It is also very true that some proteins don't localize to cilia but still cause ciliopathies, like GANAB and ALG5. Gpr161 is considered a ciliary GPCR but it also carries cilium-independent functions. Also, there is no evidence so far demonstrating that INPP5E has no extra-ciliary functions. What matters is whether the

mutation/disease phenotypes are caused with defective cilia structure and/or function.

With that said, to my understanding, the novel findings of current manuscript are 1) the ciliary PIP3 at the transition zone is regulated by PI3Ka and PI3Kb, 2) PI3Ka activity changes the phosphorylation of CEP170, and 3) the CEP170-KIF2A axis may be the ciliary effector of PI3Ka activation to promote cilia disassembly. Would the authors agree that these are direct effects on cilia?

Reviewer #3 (Remarks to the Author):

The authors have addressed my concerns.

Reviewer #3 (Remarks to the Author):

The authors have addressed my concerns.

Reviewer #1 (Remarks to the Author):

Overall, the authors have comprehensively responded to my concerns and there are no major issues remaining that would preclude publication in my opinion. I do have a couple of follow up points to consider in terms of data presentation which may still improve the manuscript:

(1) The reviewer appreciated the inclusion of the full stats table in the reviewer response. I believe this would be beneficial to include in the supplement, since it fully supports the conclusions (though I concede the authors' point that the comparisons are too numerous for inclusion in the figure).

We have included the full Tukey's post-hoc analysis table for Fig. 7h in the revised manuscript (Supplementary Table 3).

(3) The reviewer appreciated the author's plotting the data from figures 4a and c as a super plot, which does alleviate the concerns about the robustness of the data. I would recommend using these super-plots in place of the histograms as they are much more informative. The author's are correct that showing the means and SD is not appropriate for non-Gaussian distributions, but there is no requirement to use mean and SD for super plots. Indeed, a statistical analysis was performed using Kruskal-Wallis, which compares medians. The super-plot could easily compare medians of each experiment and accomplish the same effect. Although showing mean \pm SD of medians would be ok (the variance between experiment medians is likely more gaussian), medians and 95% CI of the median could also be plotted.

We are pleased to see the Reviewer's concern regarding the robustness of the data in figures 4a and c has been alleviated by plotting the data as SuperPlots. However, we maintain that the data is better presented as a histogram as we have done previously for ciliary phosphoinositide MFI data¹, as it shows the variability at the single cell level and the shift in distribution.

However, to reassure readers of the data robustness, and in reply to Reviewer 1, we now also present the data from Fig. 4a as a SuperPlot using median and 95% confidence intervals as summary statistics in the supplementary figures (Supplementary Fig. 3).

Reviewer #2 (Remarks to the Author):

This reviewer is pleased by the authors' efforts on putting together a very organized, detailed, and clear responses to reviewers' comments. Here is the remaining concern, which I hope can be addressed better by carefully constructing the introduction and discussion.

The authors' argument about the 'first-order ciliopathies' and 'second-order ciliopathies' is confusing. In the review article from Lovera et al., 'first-order ciliopathies' refer to diseases caused by gene mutations directly affect the assembly, maintenance, or function of centrioles or cilia; and 'second-order ciliopathies', also termed as 'disorders with ciliary contribution', are caused by genes with indirect effects

on cilia. Do the authors conclude their findings with PI3Ka/b direct effects or indirect effects on cilia? This needs to be clarified. To a broader interest, whether it's appropriate to call cancers 'disorders with ciliary contribution' would be a question remain to be discussed.

I agree with the authors that many "cilium-specific" proteins are later found with non-ciliary localization and function and it is true vice versa. It is also very true that some proteins don't localize to cilia but still cause ciliopathies, like GANAB and ALG5. Gpr161 is considered a ciliary GPCR but it also carries cilium-independent functions. Also, there is no evidence so far demonstrating that INPP5E has no extra-ciliary functions. What matters is whether the mutation/disease phenotypes are caused with defective cilia structure and/or function.

With that said, to my understanding, the novel findings of current manuscript are 1) the ciliary PIP3 at the transition zone is regulated by PI3Ka and PI3Kb, 2) PI3Ka activity changes the phosphorylation of CEP170, and 3) the CEP170-KIF2A axis may be the ciliary effector of PI3Ka activation to promote cilia disassembly. Would the authors agree that these are direct effects on cilia?

We agree with the last 3 points mentioned by Reviewer 2, including that the effects of PI3K on primary cilia biology is via a novel PIP₃/PDK1/PCK1/CEP170/KIF2A signalling axis.

The Reviewer also highlighted inconsistencies in the interpretation by different groups of 'direct' vs 'indirect' effects on primary cilia, as well as to the definition of '*disorders with cilia contributions*'. To address these issues, and in reply to Editor feedback to provide conceptual discussion and clarification in our revised manuscript, we have now updated the text, including a new discussion paragraph with a critical analysis of the literature related to these definitions, and how our findings integrate with this wider field.

A first question revolves around the definition of a 'direct' effect on cilia. PI3K acts by phosphorylating a membrane lipid in cilia which then modulates the function of downstream effector proteins that impact cilia biology. PI3K could therefore be defined as having an 'indirect' effect on cilia. However, one could also argue for a 'direct' effect of PI3K on cilia, given that the PI3K lipid substrates and effector proteins localise to the cilium. Therefore, in the discussion we propose that the field should not use the terms direct/indirect with regards to cilia regulators, given that (1) this definition is open to interpretation and – importantly – (2) the emerging continuum of key cilia mediators that do not fit such a narrow dichotomous classification.

Secondly, there also appears to be some uncertainty regarding the definitions of '*first*' and '*second*' '*order ciliopathies*' in relation to the definition of '*disorders with cilia contributions*'. The definition of 'first' and 'second' order ciliopathies does not address whether the mutations have a direct or indirect effect on primary cilia function, and only refers to the localisation of the mutant proteins (i.e. localising to cilia/centrosomes or not, respectively).

To address these wider issues, we have made the following changes to the manuscript:

- Revised introduction: inclusion of definitions of 'first' and 'second' '*order ciliopathies*' (Page 2, Line 66-69)
- Updated discussion:
 - o Clarification that we do not draw a conclusion on 'first' and 'second' '*order ciliopathies*' given that we are unable to determine the sub-cellular localisation of PI3K isoforms due to the lack of appropriate tools (Page 10, Line 504-505).

- Addition of 3 paragraphs critically assessing the literature on the use of 'direct/indirect effect' on cilia and examining how the definition of '*disorders with cilia contributions*' relates to our findings for *PIK3CA* mutant diseases and models (Page 11, Line 559-581).

Specifically with regards to the Reviewer comment on INPP5E, we would like to note that in addition to localisation and function at the cilium, this phosphatase also localises to the cytosol with Golgi enrichment² and has been shown to regulate autophagy³, phagocytosis⁴, apoptosis⁵ and insulin signalling⁶, although these functions have not been directly related to the *in vivo* human or mouse *INPP5E* mutant phenotypes. Therefore, there is significant evidence showing INPP5E has multiple extra-ciliary functions.

References

1. Conduit SE, *et al.* A compartmentalized phosphoinositide signaling axis at cilia is regulated by INPP5E to maintain cilia and promote Sonic Hedgehog medulloblastoma. *Oncogene* **36**, 5969-5984 (2017).
2. Kong AM, *et al.* Cloning and characterization of a 72-kDa inositol-polyphosphate 5-phosphatase localized to the Golgi network. *J Biol Chem* **275**, 24052-24064 (2000).
3. Hasegawa J, Iwamoto R, Otomo T, Nezu A, Hamasaki M, Yoshimori T. Autophagosome-lysosome fusion in neurons requires INPP5E, a protein associated with Joubert syndrome. *EMBO J* **35**, 1853-1867 (2016).
4. Horan KA, *et al.* Regulation of FcγR-stimulated phagocytosis by the 72-kDa inositol polyphosphate 5-phosphatase: SHIP1, but not the 72-kDa 5-phosphatase, regulates complement receptor 3 mediated phagocytosis by differential recruitment of these 5-phosphatases to the phagocytic cup. *Blood* **110**, 4480-4491 (2007).
5. Kisseleva MV, Cao L, Majerus PW. Phosphoinositide-specific inositol polyphosphate 5-phosphatase IV inhibits Akt/protein kinase B phosphorylation and leads to apoptotic cell death. *J Biol Chem* **277**, 6266-6272 (2002).
6. Bertelli DF, *et al.* Phosphoinositide-specific inositol polyphosphate 5-phosphatase IV inhibits inositol trisphosphate accumulation in hypothalamus and regulates food intake and body weight. *Endocrinology* **147**, 5385-5399 (2006).

REVIEWERS' COMMENTS

Reviewer #2 (Remarks to the Author):

I don't have further questions.